# TIME-ACCURATE SPEECH RICH TRANSCRIPTION WITH NON-FLUENCIES

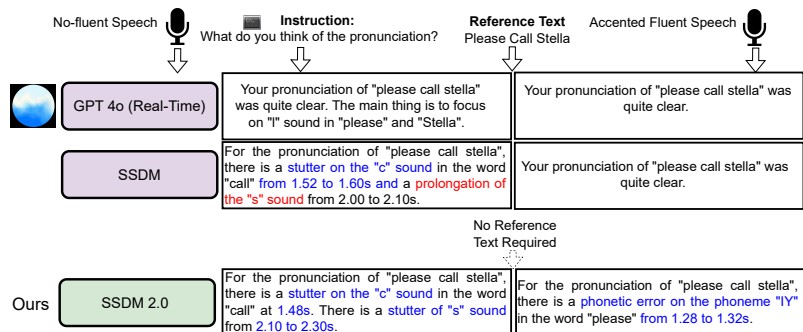

Figure 1: SSDM 2.0 Demo. Audios available at `https://shorturl.at/ITUu0`.

## ABSTRACT

Speech is a hierarchical collection of text, prosody, emotions, dysfluencies, etc. Automatic transcription of speech that goes beyond text (words) is an underexplored problem. We focus on transcribing speech along with non-fluencies (dysfluencies). The current state-of-the-art pipeline (Lian et al., 2024) suffers from complex architecture design, training complexity, and significant shortcomings in the local sequence aligner, and it does not explore in-context learning capacity. In this work, we propose SSDM 2.0, which tackles those shortcomings via four main contributions: (1) We propose a novel *neural articulatory flow* to derive highly scalable speech representations. (2) We developed a *full-stack connectionist subsequence aligner* that captures all types of dysfluencies. (3) We introduced a mispronunciation prompt pipeline and consistency learning module into LLM to leverage dysfluency *in-context pronunciation learning* abilities. (4) We curated Libri-Dys (Lian et al., 2024) and open-sourced the current largest-scale co-dysfluency corpus, *Libri-Co-Dys*, for future research endeavors. In clinical experiments on pathological speech transcription, we tested SSDM 2.0 using nfvPPA corpus primarily characterized by *articulatory dysfluencies*. Overall, SSDM 2.0 outperforms SSDM and all other dysfluency transcription models by a large margin. See our project demo page at `https://srnf2.github.io/`.

## 1 INTRODUCTION

Current automatic speech recognition (ASR) systems (Radford et al., 2023) and speech language models (SLMs) (Wu et al., 2024) typically transcribe speech into *the words that were spoken (lexicalized speech)* rather than *how the words were spoken (uttered speech)*. For example, when someone says *P-Please c-call st-ah-lla*, these systems usually perform *denoising* and output *please call stella*. This approach is suitable for most spoken dialogue scenarios and services, and helps reduce confusion in communications. However, in *pathological speech* domain, *uttered speech* is required to accurately identify articulation and pronunciation problems for diagnostic purposes. The gap between lexicalized speech and uttered speech is referred to as *dysfluency*, which includes *repetition, deletion, insertion, replacement of sounds, filler words, and hesitations* (Lian et al., 2023b). Accu-

---

[1]Terms *Dysfluency, Non-fluency, dysfluency* are interchangeable.

rate transcription of speech dysfluencies could substantially reduce the workload of speech language pathologists while facilitating diagnosis and serving as a powerful clinical assessment tool.

Pathological speech disorders are typically caused by neurological or physiological factors and are associated with various dysfluencies, such as motor and phonological (articulatory) dysfluencies (e.g., nfvPPA, Parkinson's disease, Broca's aphasia) and higher-order (or semantic) dysfluencies (e.g., svPPA, Wernicke's aphasia, ASD). Diagnosing these disorders is challenging due to case-by-case variability and differences in severity. However, they share a common set of dysfluencies at the behavioral level, making it possible to develop a general dysfluency transcription system that can accommodate all disorders and support follow-up diagnosis. Due to data constraints, however, it is not feasible to test such a system on all pathological speech data. Therefore, we focus on nfvPPA, leveraging the currently available data to develop a dysfluency transcription tool specifically targeting *articulation-based dysfluencies*.

Technically, SSDM (Lian et al., 2024) is the first end-to-end pipeline that can transcribe both *lexicalized speech* and *uttered speech (with dysfluencies)*. However, it faces challenges in representation learning complexity, limited dysfluency alignment coverage, and minimal performance boost from language modeling. Technically, there are four questions to address: (1) What are the most scalable speech representations? (2) How to align dysfluent speech with text? (3) How to curate large-scale dysfluent speech data with time-aware annotations? (4) How to leverage pronunciation in-context learning from large language models? We aim to address the aforementioned challenges and present **SSDM 2.0**. Our key contributions are as follows:

- We propose *Neural Articulatory Flow*, which encodes a *semi-implicit speech representation* that has been shown to be the most scalable dysfluency-aware representation, inspired by *articulatory dysfluencies* in pathological speech disorders.
- We develop *Full-Stack Connectionist Subsequence Aligner (FCSA)*, achieving comprehensive coverage of dysfluency alignments and precise distribution estimation.
- We curate and opensource *Libri-Dys-Co*, the largest simulated speech co-dysfluency corpus, featuring over 6,000 hours of time-aware dysfluency annotations (word/phoneme).
- We introduce *Mispronunciation In-Context Learning* and *Consistency Learning* in langauge model to achieve zero-shot dysfluency transfer and joint fluent-dysfluent ASR tasks transfer.

SSDM 2.0 significantly and consistently outperforms all existing methods, including speech language models, and can serve as a foundational dysfluency modeling tool.

## 2 SSDM 2.0 OVERVIEW

SSDM 2.0 (shown in Fig. 2) processes dysfluent speech and a textual prompt as inputs, generating pronunciation transcriptions as output. To achieve this, we propose the *Neural Articulatory Flow (NAF)*, which generates scalable speech representations referred to as *gestural scores*. Subsequently, the gestural scores are aligned with the reference text using a *Full-stack Connectionist Subsequence Aligner*, producing aligned speech embeddings for each token in the reference text. These aligned embeddings, combined with pre-defined prompts, are then input into a LLaMA (Touvron et al., 2023) module for instruction tuning (Gong et al., 2023b), a process we term *Non-fluency In-context Learning*. The following subsections provide a detailed explanation of each module.

## 3 NEURAL ARTICULATORY FLOW

Current speech representation modeling typically uses explicit $D \times T$ matrices, where T represents time and D represents channel dimension, and these are learned densely in a data-driven manner. However, human speech is produced by a sparse set of articulators with sparse activation in time. If we define basic moving patterns of articulators as a dictionary and project articulatory data into this dictionary space, we obtain a sparse activation matrix. The dictionary is called *gestures* and the sparse activation matrix is called *gestural scores* (Browman & Goldstein, 1992). This motivates us to ask: instead of densely learning elementwise speech representations, can we sparsely and implicitly learn such structural speech representations via human-prior rules? At the same time, as we focus on *articulatory dysfluencies*, modeling speech production processes helps identify the specific articulation problems in the gestural space and brings the representation closer to real human dysfluent speech. In this work, we propose *Neural Articulatory Flow*, which involves a *Semi-*

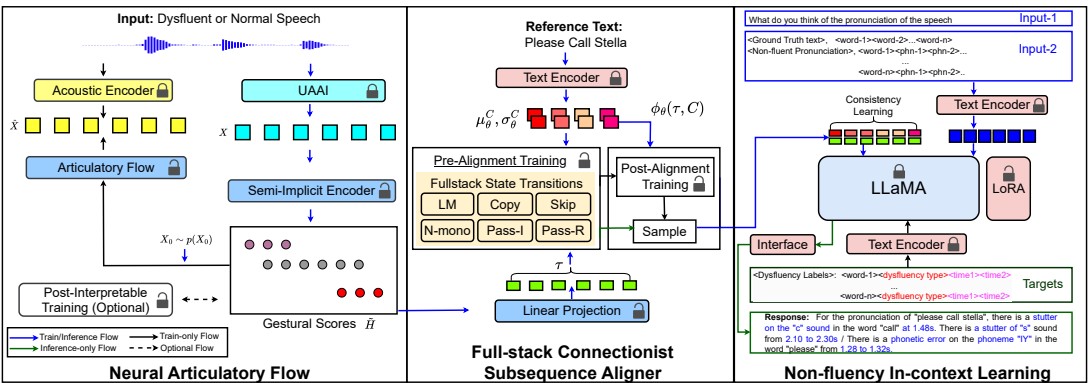

Figure 2: SSDM 2.0 architecture

*implicit Speech Encoder* (Section 3.1) to predict only the indices of active regions in articulation, and an *Articulatory Flow* (Section 3.2) to distill speech intelligibility from pretrained acoustic speech embeddings (WavLM (Chen et al., 2022)). We also optionally introduce *Interpretable Posterior Training* to visualize how speech is physically produced in articulation and to derive articulatory feedback for pronunciation.

## 3.1 SEMI-IMPLICIT SPEECH ENCODER

Given a speech waveform, we adopt UAAI (**U**niversal **A**coustic-to-**A**rticulatory **I**Inversion)(Cho et al., 2024) to obtain its articulatory trajectory $X \in \mathbb{R}^{D \times T}$ at 50Hz. UAAI consists of a WavLM(Chen et al., 2022) encoder followed by a linear layer, which takes speech waveform as input and predicts articulatory trajectories. The model was trained on the MNGU0 corpus (Richmond et al., 2011), which contains 75 minutes of EMA (Electromagnetic midsagittal articulography) data collected during newspaper reading. In our setting, $D = 12$ corresponds to the x and y coordinates of 6 articulators measured in the EMA recording process.

We define articulatory trajectory kernels (*gestures*) $G^{T' \times D \times K}$, where $T'$ is the window size and $K$ is the number of kernels. By projecting $X$ onto the kernel space, we obtain *gestural scores* $H \in \mathbb{R}^{K \times T}$, which is high-level abstraction of $X$. $H$ denotes when the articulators are activated for how long. In this work, we directly predict $H$ from $X$ without *gestures* $G$ for simplicity, focusing only on active indices that indicate articulatory activation with a *Count Encoder*, a *Index Encoder* and a *Value Encoder*. We provide a visualization of gestural scores $H$ and its correlation with $X$ and *Gestures* $G$ in Appendix. A.5.

In our implementation, as shown in Fig. 3, the *Count Encoder* projects $X$ into a $K \times T$ matrix. Each row $X_i \in \mathbb{R}^T$ is projected into a discrete number $Z_{C_i}$ sampled from $q_\theta(Z_{C_i}|X_i)$, indicating the number of activation regions. An *Index Generator* takes $X_i$ as input to generate two discrete numbers, repeated $Z_{C_i}$ times. After sorting, this yields $Z_{I_i} = \left[ (Z_{I_i}^{2\tau-1}, Z_{I_i}^{2\tau}) \right]_{\tau=1}^{Z_{C_i}}$ where $(Z_{I_i}^{2\tau-1}, Z_{I_i}^{2\tau})$ are start and end indices of the $\tau$-th region in row $i$. An *Index Compiler* predicts values for each span $(Z_{I_i}^{2\tau-1}, Z_{I_i}^{2\tau})$ in row $i$, returning $X_i^\tau = X_i \left[ :, Z_{I_i}^{2\tau-1} \text{ to } Z_{I_i}^{2\tau} \right]$. A *Value Encoder* then predicts continuous values for $H_i^\tau = H_i \left[ :, Z_{I_i}^{2\tau-1} \text{ to } Z_{I_i}^{2\tau} \right]$. Since $H$ is implicitly defined by durations and values, yet maintains structural properties like sparsity typical of explicit representations, we call it *Semi-implicit Representation* and the module the *Semi-implicit Encoder*.

**Posteriors and Priors**    $Z_{C_i}$ and $(Z_{I_i}^{2\tau-1}, Z_{I_i}^{2\tau})$ are discrete variables. We set prior $p(Z_{C_i}) = 1/\mathbb{C}1$, and $p(Z_{I_i}^{2\tau}) = p(Z_{I_i}^{2\tau-1}) = 1/\mathbb{C}_2$, where $\mathbb{C}_1$ is the maximum number of spans, $\mathbb{C}_2$ is the maximum end index. We define $\mathbb{C}_1 = T/4$ and $\mathbb{C}_2 = T$, where $T$ is the total number of timesteps at 50Hz. Following Lian et al. (2024), we adopt Gumbel-Softmax (Jang et al., 2016) for the discrete posterior. The posterior for $Z_{C_i}$ and $(Z_{I_i}^{2\tau-1}, Z_{I_i}^{2\tau})$ is formulated in Eq. 1, where $\tilde{\pi}_c^i = (\log(\pi_c^i) + \epsilon_c^i)/\varsigma$, $c$ and $l$ are discrete label class indices, $\varsigma$ is the temperature parameter, and Gumbel noise $\epsilon_c^i = -\log(-\log(U_c))$ where $U_c \sim \text{Uniform}(0,1)$. $\pi^i \in \mathbb{R}^{\mathbb{C}_1}$ is probability logits over $\mathbb{C}_1$ classes. $\tilde{\pi}_c^{i,\tau}$ is defined under the same criterion with one additional span index $\tau$. For the continuous posterior

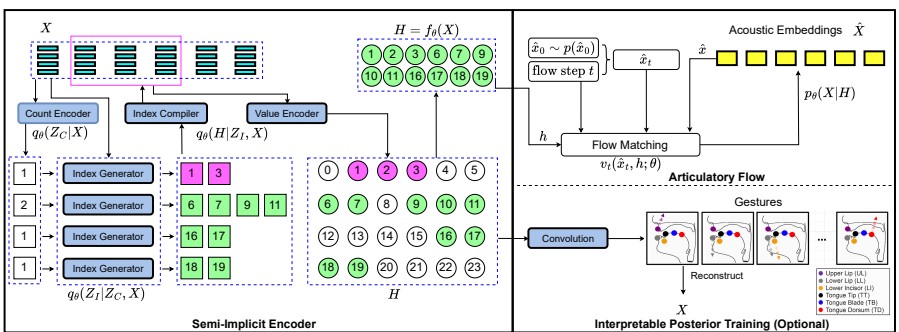

Figure 3: Neural Articulatory Flow operates as follows: For a $4 \times 6$ matrix $H$, the Counter Encoder generates [1,2,1,1], denoting active regions per row. The Index Generator predicts start and end indices, e.g., [1,3] for row 1, indicating non-zero entries. $X[:, 1:3]$ then predicts three continuous values for $H$. $H$, being rule-generated and sparse, represents a semi-implicit representation. It generates speech $\hat{X}$ for intelligibility, followed by post-interpretable training to enhance interpretability.

$q_\theta(H_i^\tau | Z_{I_i}^{2\tau-1}, Z_{I_i}^{2\tau}, X_i)$, we set $p(H_i^\tau) \sim \mathcal{N}(0, I)$.

$$q_\theta(Z_{C_i}=c|X_i) \approx \frac{\exp\left(\tilde{\pi}_c^i\right)}{\sum_{l=1}^{\mathbb{C}_1} \exp\left(\tilde{\pi}_l^i\right)}, \quad q_\theta(Z_{I_i}^{2\tau \text{ or } 2\tau-1}=c|X_i, Z_{C_i}) \approx \frac{\exp\left(\tilde{\pi}_c^{i,\tau}\right)}{\sum_{l=1}^{\mathbb{C}_2} \exp\left(\tilde{\pi}_l^{i,\tau}\right)} \quad (1)$$

**KL Loss** The *Count Encoder* models $q_\theta(Z_C|X) = \prod_{i=1}^{K} q_\theta(Z_{C_i}|X_i)$. The *Index Generator* models $q_\theta(Z_I|X, Z_C) = \prod_{i=1}^{K} \prod_{\tau=1}^{Z_{C_i}} q_\theta(Z_{I_i}^{2\tau-1}|X_i, Z_{C_i}) q_\theta(Z_{I_i}^{2\tau}|X_i, Z_{C_i})$. The *Index Compiler* together with *Value Encoder* models $q_\theta(H|Z_I, X) = \prod_{i=1}^{K} \prod_{\tau=1}^{Z_{C_i}} q_\theta(H_i^\tau | Z_{I_i}^{2\tau-1}, Z_{I_i}^{2\tau}, X_i)$. The joint priors $p(Z_C) = \prod_{i=1}^{K} p(Z_{C_i})$, $p(Z_I) = \prod_{i=1}^{K} \prod_{\tau=1}^{Z_{C_i}} p(Z_{I_i}^{2\tau-1}) p(Z_{I_i}^{2\tau})$, $p(H) = \prod_{i=1}^{K} \prod_{\tau=1}^{Z_{C_i}} p(H_i^\tau)$. The KL loss is displayed in Eq. 2.

$$\mathcal{L}_{\text{KL}} = \mathbb{E}_{X \sim p(X)}[\text{KL}(q_\theta(Z_C \,|\, X) \,\|\, p(Z_C)) + \text{KL}(q_\theta(Z_I \,|\, X, Z_C) \,\|\, p(Z_I)) + \text{KL}(q_\theta(H \,|\, Z_I, X) \,\|\, p(H))] \quad (2)$$

## 3.2 ARTICULATORY FLOW

Articulatory flow simulates the human speech production process. Given sparse gestural scores $H$, each row vector $H_i$ controls the movement of one of $K$ articulators (Lian et al., 2024). While early work achieved articulatory synthesis using articulatory kinematics data, we take a different approach. We generate speech (WavLM (Chen et al., 2022) representations) $\hat{X}$ directly from sparse gestural scores $H$ using conditional flow matching (CNF) (Lipman et al., 2022). CNF models a vector field $v_t : [0, 1] \times \mathbb{R}^K \to \mathbb{R}^D$ that constructs the flow $\phi_t : [0, 1] \times \mathbb{R}^K \to \mathbb{R}^D$ that maps priors to speech representations, satisfying $\frac{d}{dt}\phi_t(\hat{x}) = v_t(\phi_t(\hat{x}))$; $\phi_0(\hat{x}) = \hat{x}$, where $t$ is the step index and $\hat{x} \in \mathbb{R}^D$ is a column vector of $\hat{X}$. We follow Le et al. (2024) for vector field implementation. Specifically, at each step $t$, noise $\hat{x}_0$ is sampled, which gives $\hat{x}_t = (1 - (1 - \sigma_{\min})t)\hat{x}_0 + t\hat{x}$. We set $u_t(\hat{x}_t|\hat{x}) = \hat{x} - (1 - \sigma_{min})\hat{x}_0$. A sinusoidal position embedder is employed for step encoding, denoted as $s_t \in \mathbb{R}^K$, where $K$ is the number of gestures. We use matrices $\hat{X}_t, \hat{X}, H$ in the actual computation. $s_t$ is concatenated with $H \in \mathbb{R}^{K \times T}$, $\hat{X}_t$, and $\hat{X}$ to form $\tilde{H} \in \mathbb{R}^{(K+2D) \times (T+1)}$, which is then used to predict $V_t \in \mathbb{R}^{D \times T}$, which matches the dimension of $\hat{X}$. We keep the vector form in the loss objective, shown in Eq. 3. Note that we do not perform an inference step, as we only use articulatory flow to inject intelligibility into our gestural scores.

$$\mathcal{L}_{\text{FLOW}} = \mathbb{E}_{t, q(\hat{x}, h), p_0(\hat{x}_0)}\left[\|u_t(\hat{x}_t \mid \hat{x}) - v_t(\hat{x}_t, h; \theta)\|^2\right], \quad (3)$$

## 3.3 POST-INTERPRETABLE TRAINING

Gestural scores are traditionally paired with gestures (Browman & Goldstein, 1992; Ramanarayanan et al., 2013). This work removes such constraint by deriving only the former, simplifying overall

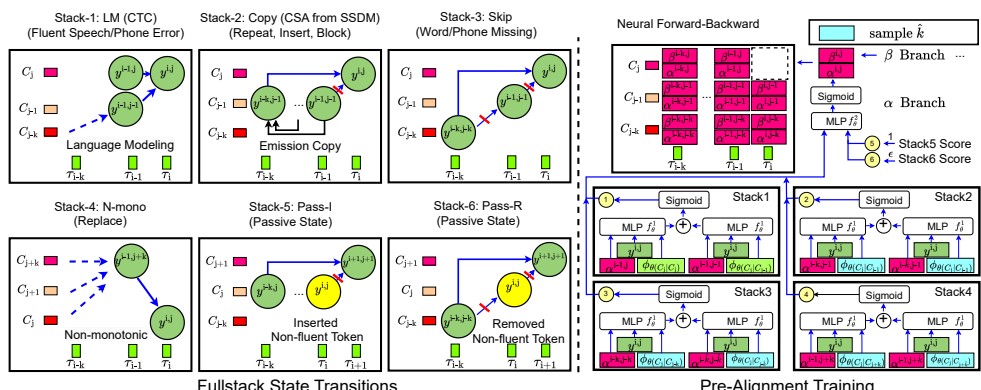

Figure 4: Fullstack Connectionist Subsequence Aligner (FCSA) employs six-stack optimization for speech transitions: four active (1-4) and two passive (5-6) states. The accompanying figure shows the neural forward algorithm; with the backward process in Appendix A.9.

system training as described in Lian et al. (2024). However, interpreting gestural scores helps locate specific *pronunciation errors* related to articulators (Lian et al., 2024). To leverage this property, we *optionally* perform post-training by applying k-means clustering to articulatory data $X$ and obtaining gestures $G \in \mathbb{R}^{T' \times 12 \times K}$, where $T'$ is the window size, 12 corresponds to $x$ and $y$ coordinates of 6 articulators, and $K$ is the number of gestures. We then employ neural convolutive matrix factorization (Lian et al., 2022) for gestural scores $H$ and gestures $G$ to reconstruct $X$, ensuring $H$ is interpretable as actual human speech production. The loss $\hat{\mathcal{L}}_{\text{PIT}}$ is the reconstruction loss for $X$ without additional sparsity constraints. Details are presented in Appendix. A.5.

## 4 FULLSTACK CONNECTIONIST SUBSEQUENCE ALIGNER

### 4.1 FULLSTACK STATE TRANSITIONS

Given frame-wise speech tokens $\tau = [\tau_i \in \mathbb{R}^D]_{i=1}^T$ and reference text tokens $\gamma(C) = [\gamma(C_j) \in \mathbb{R}^D]_{j=1}^L$, the detection of non-fluency in speech hinges on the alignment, which is usually a $L \times T$ matrix. Emission probability $y^{i,j} = p(\tau_j|\tau_i)$ and transition probability $p(C_m|C_n)$ are usually introduced for optimizing the alignment learning. For example, when the speech is normal or fluent, the alignment would be completely monotonic and $y^{i,j}$ only depends on $y^{i-1,j}$ and $y^{i-1,j-1}$, which is the case of vanilla CTC optimization (Graves et al., 2006), and we call this **Stack-1**, named as *LM(CTC)*, as shown in Fig. 4. Stack-1 encodes normal language modeling. Note that in the future discussion we use $y^{i,j}$ to refer to both emission probability and position tokens $(i, j)$ for convenience. For non-fluent speech, the alignment cases are diverse. Assume $\tau_{i-k}$ and $\tau_i$ are aligned with $C_{j-k}$ and $C_j$ respectively, and the other speech frames $[\tau_{i-k+1}, \ldots, \tau_{i-1}]$ are dysfluencies including repetition, insertion, and block. Then we only consider $y^{i-k,j-1} \to y^{i,j}$. CSA (Lian et al., 2024) implicitly achieves this by performing *Emission Copy* such that $y^{i-k,j-1} = \ldots = y^{i-1,j-1}$ so that we could still apply the normal language modeling $y^{i-1,j-1} \to y^{i,j}$. This is **Stack-2**, named as *Copy*. To give a concrete example, if speech $\tau = [P, L, IY, L, IY, Z]$ (p-l-ea-l-ea-se) and text $C = [P, L, IY, Z]$ (please), then the last $[\tau_{i-k+1}, \tau_{i-1}] = [L, IY]$ are inserted or repetition tokens. In this work, we propose *Fullstack State Transitions* in addition to these basic stacks. We introduce the other four stacks in the following. **Stack-3**, designated as the *Skip* stack, addresses missing dysfluencies. As illustrated in Fig. 4, the reference text $C_{j-1}$ is omitted, consequently skipping the emission $y^{i-1,j-1}$. In this scenario, the transition is represented as $y^{i-k,j-k} \to y^{i,j}$, which constitutes a *disrupted language model*. **Stack-4**, termed *N-mono*, introduces non-monotonicity that may indicate replacement errors. For instance, the pronunciation $[P, L, EY, Z]$ in contrast to $[P, L, IY, Z]$ for the word *please*. While Stacks 1-4 primarily focus on fluent tokens $y^{i,j}$, where speech $\tau_i$ precisely aligns with text $C_j$, misalignments can occur, such as $y^{i-1,j-1}$ in Stack-2 and Stack-3. To address these cases, we introduce **Stack-5**, wherein $y^{i,j}$ represents a passive state indicating that $\tau_i$ is an *inserted non-fluent token*. We designate this as *Pass-I*. Similarly, **Stack-6** incorporates a passive state corresponding to a *removed non-fluent token*, which we denote as *Pass-R*. Based on these

full-stack state transitions, we propose two novel training approaches: *Pre-Alignment Training* and *Post-Alignment Training*. The former incorporates a neural forward-backward algorithm, while the latter is designed to stochastically optimize the non-fluent full-stack alignments. We call our method *Fullstack Connectionist Subsequence Aligner*. More context is provided in Appendix. A.7 and A.8.

## 4.2 PRE-ALIGNMENT TRAINING

Define alignment function $\gamma(C, \tau) = [\gamma(C_j)]_{j=1}^L$ such that $\gamma(C_j, \tau) = [\tau_{s_j}, \tau_{e_j}]$ where: $1 \le s_j \le e_j \le T, e_j \le s_{j+1}, s_j < s_{j+1}, e_j < e_{j+1}$ for all $j \in 1, 2, \ldots, L-1$. This formulation ensures that all elements $\tau_i$ in $H$ are uniquely aligned to a target text token. It is important to note that $\gamma(C_j, \tau)$ may be an empty set, indicating that the corresponding text is absent from the speech. There are multiple possible alignments for each speech-text pair. Thus, we define $\Gamma(C, \tau) = \gamma_i(C, \tau)_{i=1}^N$ to represent all $N$ possible alignments. We aim to find a stochastic alignment $\Gamma\theta(C, \tau)$ that encompasses all six stacks. In this case, the objective can be simply expressed as in Eq. 4.

$$\max_\theta \mathbb{E}_{C,\tau} \left[ p_\theta(\Gamma(C, \tau)) \right] = \max_\theta \mathbb{E}_{C,\tau} \left[ \sum_{i=1}^N p_\theta(\gamma_i(C, \tau)) \right] \tag{4}$$

Note that CTC (Graves et al., 2006) (Stack-1 only) represents a special case of this formulation when only monotonic alignment is considered. In this context, the forward-backward algorithm is utilized to model $p_\theta(\gamma_i(C, \tau))$. For joint Stack1-Stack2 probabilistic modeling, an LCS-aware (Hirschberg, 1977) forward-backward algorithm (Lian et al., 2024) is employed. In this work, we propose the *Neural Forward-Backward Algorithm* (NFB) to model $p_\theta(\gamma_i(C, \tau))$ across all six stacks (Stack 1-6). We start with deriving the emission probability $y^{i,j} = p_\theta(C_j|\tau_i) \approx \exp(\tau_i \cdot C_j^S) / \sum_{k=1}^L \exp(\tau_i \cdot C_k^S)$, where $C_j^S$ is sampled from $\mathcal{N}(\mu_\theta^{C_j}, (\sigma_\theta^{C_j})^2)$, which is modeled by the text encoder (Lian et al., 2024). Let us examine the transition dependencies in the forward branch ($\alpha$ branch). $\alpha^{i,j}$ is derived from one or two previous states, as specified in the six stacks. The challenge lies in the uncertainty regarding the actual distribution of dysfluencies in speech at the frame level, precluding the simple application of decayed hyperparameters for these stacks, as proposed by Lian et al. (2024). To address this, we introduce a simple multi-layer perceptron module that takes $\alpha^{m,n}$ and the corresponding transition /emission probability as input. The outputs are summed and processed by a sigmoid function ($f^0$) to produce a score. We denote this MLP module as $f_\theta^1$, which is shared across all stacks. Let $\alpha_u^{i,j}$ represent the score output from Stack-$u$. The following rules then apply:

$$\alpha_1^{i,j} = f^0 \left( f_\theta^1(\alpha^{(i-1,j)}, \phi_\theta(C_j|C_j), y^{i,j}) + f_\theta^1(\alpha^{(i-1,j-1)}, \phi_\theta(C_j|C_{j-1}), y^{i,j}) \right) \tag{5}$$

Stacks 2-4 correspond to the non-fluency forward process, which is uniformly shown in Eq. 6.

$$\alpha_u^{i,j} = f^0 \left( f_\theta^1(\alpha^{(i-a_u,j-b_u)}, \phi_\theta(C_j|C_{j-b_u}), y^{i,j}) + f_\theta^1(\alpha^{(i-\hat{a}_u,j-\hat{b}_u)}, \phi_\theta(C_j|C_{j-\hat{b}_u}), y^{i,j}) \right) \tag{6}$$

where $u \in \{2, 3, 4\}, (a_2, b_2) = (k, 1), (\hat{a}_2, \hat{b}_2) = (\hat{k}, 1), (a_3, b_3) = (k, k), (\hat{a}_3, \hat{b}_3) = (\hat{k}, \hat{k}), (a_4, b_4) = (1, -k), (\hat{a}_4, \hat{b}_4) = (1, -\hat{k})$, and $k \le \hat{k} \le \min(i, j) - 1$ are randomly sampled to increase dysfluency diversity. For the passive states (stacks 5 and 6), we set $\alpha_5^{i,j} = 1$ and $\alpha_6^{i,j} = \epsilon = 10^{-5}$. The intuition behind this is that an inserted non-fluent token has no influence on future states $\alpha^{i+1,j+1}$ but maintains the history $y^{i-k,j} \to y^{i,j}$. Conversely, a removed non-fluent token severs the information flow, as $y^{i,j}$ is detached from both $y^{i+1,j+1}$ and $y^{i-k,j-k}$. Introducing another MLP module $f_\theta^2$ and employing the same sigmoid function $f^0$, we obtain:

$$\alpha^{i,j} = f^0 \left( f_\theta^2 \left( \sum_{u=1}^6 \alpha_u^{i,j} \right) \right) \tag{7}$$

We obtain $\beta^{i,j}$ similarly, as detailed in Appendix A.9. Our proposed fullstack connectionist subsequence aligner (FCSA) loss objective is shown in Eq.8. Following Graves et al. (2006), we initialize $\alpha^{1,1} = \beta^{-1,-1} = 1, \beta(:, 1) = \alpha(:, 1) = 0$, where $-1$ denotes the last token index. During next stage, we adopt the longest common subsequence (Lian et al., 2024) for sampling the alignment.

$$\mathcal{L}_{\text{PRE}} = -\mathbb{E}_{C,\tau} \left[ \sum_{i=1}^N p_\theta(\gamma_i(C, \tau)) \right] = -\mathbb{E}_{C,\tau,i,j} \left[ \frac{\alpha^{i,j} \beta^{i,j}}{y^{i,j}} \right] \tag{8}$$

### 4.3 POST-ALIGNMENT TRAINING

The application of *Pre-Alignment Training* is predicated on the availability of only clean text and non-fluent (or noisy) speech, which inherently lack natural monotonic alignment. However, our data simulation stage provides access to ground truth non-fluent text (phonemes), enabling the implementation of additional training paradigms. We utilize speech input $\tau = [\tau_i]_{i=1}^T$ in conjunction with non-fluent text tokens $C^{\text{NF}} = [C_i^{\text{NF}}]_{i=1}^L$. In this context, the alignment $\gamma(\tau, C^{\text{NF}})$ exhibits strict monotonicity, corresponding to *Stack-1* as illustrated in Fig. 4. For this post-training objective, we employ the vanilla CTC loss (Graves et al., 2006) function, showing in Eq. 9.

$$\mathcal{L}_{\text{POST}} = \mathbb{E}_{C^{\text{NF}}, \tau} \left[ \mathcal{L}_{\text{CTC}}(C^{\text{NF}}, \tau) \right] \tag{9}$$

## 5 NON-FLUENCY IN-CONTEXT LEARNING

### 5.1 MISPRONOUNCED PROMPT

SSDM (Lian et al., 2024) concludes that the inclusion of language models yields minimal performance improvement. We hypothesize that this is because language models may have memorized existing fluent word-phoneme mappings. For instance, when encountering a non-fluent pronunciation such as `<please><P><Block><P><L><IY><Z>`, language models tend to bias towards the fluent pronunciation `<please><P><L><IY><Z>`. To mitigate this issue, we augment the input by including all non-fluent pronunciations in the sentence. This format takes the structure `<Non-fluent Pronunciation>,<word1><phn><non-fluency><...>,` `<word2><phn><non-fluency><...>`. In addition, we include the entire word sequence, `<Ground Truth Text><word-1><word-2>...<word-n>`, to leverage zero-shot ASR performance on fluent speech. This approach aims to train the language model to recognize imperfect speech patterns. We utilize mispronounced prompts to explore zero-shot non-fluency detection performance, i.e., *non-fluency in-context learning*. Does this method improve the detection of other unseen types, such as insertion dysfluencies, even when only prompting for repetition dysfluencies?

### 5.2 CONSISTENCY LEARNING

*FCSA* (Section 4) incorporates imperfect phonetic language modeling. As described in *Mispronounced Prompts*, both fluent and non-fluent phonemes are paired with fluent words. We introduce *Consistency Learning*. Given the non-fluent speech text alignment $\gamma(C, \tau) = [\gamma(C_j)]_{j=1}^L$, we propose to align each phoneme semantically with its associated word, as presented in Eq. 10.

$$\mathcal{L}_{\text{CON}} = \sum_{j=1}^L \mathbb{E}_{\tau_j \sim \gamma(C_j)} \frac{\exp^{\tau_j^T C_j}}{\sum_{i=1, i \notin \gamma(C_j)}^T \exp^{\tau_i^T C_j}} \tag{10}$$

### 5.3 INPUT, TARGETS, TIME MODELING, LOSS OBJECTIVE

We adopt the same language model configuration as Lian et al. (2024); Gong et al. (2023b) for instruction tuning. During training, for each sample $i$, the input includes a non-fluent speech text alignment $\gamma(C^i, \tau^i)$ sampled from $p_\theta(\Gamma(C^i, \tau^i))$. The prompts comprises a general prompt such as `<what><do><you><think><of><the><pronunciation><of><the><speech>` (Input-1 in Figure 2), the ground truth word tokens, and the mispronounced prompts (Section 5.1, Input-2 in Figure 2). Subsequently, a text encoder (Gong et al., 2023b) processes these inputs. The targets are derived from our automatic annotations generated during simulation (Appendix A.1). They follow the format: `<dysfluency labels><word-1><dysfluency type><time1><time2>...<word-n><...>`, and are processed by the same text encoder. For time modeling, we adopt the frame-wise approach proposed by Huang et al. (2024). During inference, only the speech-text alignment and a general prompt (Input-1) are required. We have also developed additional interface prompts to refine the final output. The final objective is presented in Eq. 11, where $\hat{\mathcal{L}}_{\text{PIT}}$ is the optional post-interpretable loss (Sec. 3.3). $\lambda_1, \lambda_2, \lambda_3, \lambda_4, \lambda_5, \lambda_6$ are balancing factors. See details in Appendix. A.11.

$$\mathcal{L}_{\text{FINAL}} = \lambda_1 \mathcal{L}_{\text{KL}} + \lambda_2 \mathcal{L}_{\text{FLOW}} + \lambda_3 \mathcal{L}_{\text{PRE}} + \lambda_4 \mathcal{L}_{\text{POST}} + \lambda_5 \mathcal{L}_{\text{CON}} + \lambda_6 \hat{\mathcal{L}}_{\text{PIT}} \tag{11}$$

## 6 EXPERIMENTS

### 6.1 CO-DYSFLUENCY DATA

We scale the *Libri-Dys* (Lian et al., 2024) to create a larger co-dysfluency dataset named *Libri-Co-Dys*, with 6023.24 hours, compared to the *Libri-Dys*'s 3938.44 hours. Co-Dysfluency indicates that each utterance contains multiple instances of **single-type** dysfluency, and multiple instances of **multi-type** dysfluency. In *Libri-Co-Dys*, each utterance contains an average of 2.51 dysfluencies. To evaluate its utility, we also tested *Libri-Co-Dys*'s Word Error Rate (WER) and Phoneme Error Rate (PER) using Whisper (Radford et al., 2023) and phoneme recognition model (Li et al., 2020). Details of dysfluency simulation and evaluation are available in Appendix. A.1.1. We also evaluated other simulated data VCTK++ (Lian et al., 2023b), VCTK-TTS (Zhou et al., 2024b), VCTK-Stutter (Zhou et al., 2024b) and nfvPPA (Gorno-Tempini et al., 2011). Details are in Appendix. A.1.2.

### 6.2 EVALUATION METRICS

We evaluate phonetic transcription and alignment using framewise **F1 Score**, and Duration-Aware Phoneme Error Rate **(dPER)**. For dysfluency evaluation, besides F1 Scores, we report the time-aware Matching Score **(MS)**. We follow Lian et al. (2024) for scalability evaluation: *Scaling factors* SF1 for F1 score and SF2 for dPER(or MS) are computed as $(c-b) \times 0.3 + (b-a) \times 0.4$ for results [a, b, c] from Libri-Dys [30%, 60%, 100%] ( Training Data ). Details are in Appendix A.3.

Table 1: Scalable Dysfluent Phonetic Transcription Evaluation on Single-Dysfluency Corpus

| Method | Eval Data | F1 (%, ↑) | dPER (%, ↓) | F1 (%, ↑) | dPER (%, ↓) | F1 (%, ↑) | dPER (%, ↓) | F1 (%, ↑) | dPER (%, ↓) | F1 (%, ↑) | dPER (%, ↓) | SF1 (%, ↑) | SF2 (%, ↓) |
|---|---|---|---|---|---|---|---|---|---|---|---|---|---|
| Training Data | | VCTK++ | | LibriTTS (100%) | | Libri-Dys (30%) | | Libri-Dys (60%) | | Libri-Dys (100%) | | | |
| HuBERT-Large (Hsu et al., 2021) | VCTK++ | 90.5 | 40.3 | 90.0 | 40.0 | 89.8 | 41.2 | 91.0 | 40.2 | 89.9 | 41.2 | 0.15 | -0.1 |
| | Libri-Dys | 86.2 | 50.3 | 88.2 | 47.4 | 87.2 | 42.3 | 87.2 | 43.4 | 87.8 | 42.9 | 0.18 | 0.29 |
| WavLM-Large (Chen et al., 2022) | VCTK++ | 90.8 | 40.5 | 90.2 | 40.3 | 90.1 | 41.6 | 91.3 | 40.6 | 90.2 | 41.5 | 0.15 | -0.67 |
| | Libri-Dys | 86.5 | 50.7 | 88.5 | 47.8 | 87.6 | 42.7 | 87.5 | 43.7 | 88.1 | 43.2 | 0.14 | 0.25 |
| SSDM (Lian et al., 2024) | VCTK++ | 91.5 | 39.0 | 91.7 | 38.3 | 91.7 | 38.6 | 92.1 | 37.0 | 93.0 | 37.0 | 0.43 | -0.64 |
| | Libri-Dys | 88.2 | 40.9 | 88.9 | 40.9 | 89.0 | 40.8 | 89.2 | 39.0 | 90.8 | 39.0 | 0.56 | -0.72 |
| NAF w/o AF (Ours) | VCTK++ | 90.0 | 40.1 | 91.2 | 38.8 | 91.1 | 38.8 | 91.7 | 38.1 | 92.6 | 37.2 | 0.51 | -0.55 |
| | Libri-Dys | 87.6 | 41.4 | 88.5 | 41.2 | 88.2 | 41.0 | 89.0 | 39.2 | 90.3 | 38.0 | 0.71 | -0.56 |
| NAF w/ AF (Ours) | VCTK++ | **91.8** | **38.0** | **92.8** | **38.0** | **92.4** | **37.6** | **94.1** | **36.0** | **95.0** | **34.1** | **0.95** | **-1.21** |
| | Libri-Dys | **89.7** | **38.4** | **90.2** | **38.9** | **92.3** | **37.8** | **93.7** | **36.0** | **95.8** | **33.6** | **1.19** | **-1.44** |
| NAF w/ PIT (Ours) | VCTK++ | 91.6 | 38.1 | 92.6 | 38.0 | 92.3 | 37.0 | 94.0 | 36.0 | 94.7 | 34.3 | 0.89 | -0.91 |
| | Libri-Dys | 89.4 | 38.2 | 90.1 | 39.2 | 92.0 | 37.4 | 93.2 | 36.3 | 95.0 | 34.5 | 1.02 | -0.98 |

### 6.3 NEURAL ARTICULATORY FLOW IS SCALABLE PHONETIC DYSFLUENCY TRANSCRIBER

To assess the scalability of dysfluency-aware speech representations (using Neural Articulatory Flow, or NAF), we conducted framewise phoneme classification experiments using simulated data as targets. We report both framewise F1 scores and dPER (dysfluency-aware Phoneme Error Rate) in Table 1. Scalability is evaluated based on scaling factors SF1 for F1 and SF2 for dPER. We use Libri-Dys (Lian et al., 2024) (The same test set) and VCTK++ (Lian et al., 2023b) for fair comparison. We also HuBERT-Large (Hsu et al., 2021) and WavLM-Large (Chen et al., 2022), configured at 50Hz, for the same phoneme experiments. Our results demonstrate that HuBERT and WavLM exhibit poor scalability and suboptimal F1 and dPER scores. When comparing our NAF with the gestural scores in SSDM (Lian et al., 2024), we observed that without the neural articulatory flow loss $\mathcal{L}_{\text{FLOW}}$, NAF achieves lower intelligibility but still maintains better scalability. Upon incorporating the articulatory flow loss, we immediately observed significant improvements in both F1 and dPER scores, as well as scaling factors, outperforming SSDM by a considerable margin. This improvement is consistent with our understanding of articulatory flow as the process that transfers intelligibility from speech to gestural scores. It is worth noting that post-interpretable training (PIT) does not introduce additional performance improvements, as its primary function is for visualization purposes only.

Table 2: Scalable Dysfluent Phonetic Transcription Evaluation on Co-Dysfluency Corpus

| Method | Eval Data | F1 (%, ↑) | dPER (%, ↓) | F1 (%, ↑) | dPER (%, ↓) | F1 (%, ↑) | dPER (%, ↓) | SF1 (%, ↑) | SF2 (%, ↓) |
|---|---|---|---|---|---|---|---|---|---|
| Training Data | | Libri-Dys-Co (30%) | | Libri-Dys-Co (60%) | | Libri-Dys-Co (100%) | | | |
| SSDM (Lian et al., 2024) | Libri-Dys-Co | 88.6 | 42.4 | 89.0 | 39.9 | 90.0 | 39.4 | 0.46 | -1.15 |
| NAF (Ours) | Libri-Dys-Co | **92.7** | **37.9** | **93.8** | **36.2** | **96.0** | **33.8** | **1.10** | **-1.40** |

**Co-Dysfluency Scalability** We further evaluated the dysfluency phonetic alignment scalability using our Libri-Dys-Co dataset. For comparison purposes, we implemented SSDM to generate results on this dataset. Our analysis demonstrates that the Neural Articulatory Flow (NAF) consistently outperforms SSDM's gestural scores by a significant margin, as shown in Table. 2.

Table 3: Scalable Dysfluent Detection Evaluation on Single-Dysfluency Corpus

| Method | Eval Data | F1 (%, ↑) | MS (%, ↑) | F1 (%, ↑) | MS (%, ↑) | F1 (%, ↑) | MS (%, ↑) | F1 (%, ↑) | MS (%, ↑) | F1 (%, ↑) | MS (%, ↑) | SF1 (%, ↑) | SF2 (%, ↑) |
|---|---|---|---|---|---|---|---|---|---|---|---|---|---|
| Training Data | | VCTK++ | | LibriTTS (100%) | | Libri-Dys (30%) | | Libri-Dys (60%) | | Libri-Dys (100%) | | | |
| SSDM (Lian et al., 2024) | VCTK++ | 84.8 | 64.3 | 87.8 | 68.2 | 88.5 | 69.7 | 89.0 | 69.9 | 89.2 | 70.2 | 0.26 | 0.17 |
| | Libri-Dys | 78.9 | 68.3 | 79.0 | 69.4 | 79.3 | 69.8 | 80.6 | 69.9 | 81.4 | 70.4 | 0.76 | 0.19 |
| w/o LLaMA | VCTK++ | 84.5↓ | 64.0↓ | 86.9↓ | 68.0↓ | 88.4↓ | 69.7 | 88.7↓ | 69.8↓ | 88.9↓ | 69.9↓ | 0.18 | 0.07 |
| | Libri-Dys | 78.2↓ | 68.1↓ | 78.3↓ | 69.0↓ | 78.8↓ | 69.2↓ | 79.6↓ | 69.3↓ | 80.7↓ | 70.0↓ | 0.65 | 0.25 |
| w/ Curri | VCTK++ | 85.6 | 65.1 | 87.1 | 68.5 | 88.8 | 69.9 | 89.2 | 70.2 | 90.0 | 71.9 | 0.4 | 0.63 |
| | Libri-Dys | 79.2 | 68.4 | 79.4 | 69.5 | 79.4 | 69.9 | 81.0 | 70.5 | 81.6 | 71.0 | 0.82 | 0.39 |
| SSDM+NAF (Ours) | VCTK++ | 85.0 | 64.5 | 88.1 | 68.4 | 88.7 | 70.0 | 89.4 | 70.4 | 90.4 | 71.3 | 0.58 | 0.43 |
| | Libri-Dys | 79.3 | 68.5 | 79.1 | 69.7 | 79.3 | 70.0 | 81.2 | 70.8 | 83.0 | 71.2 | 1.30 | 0.44 |
| SSDM+FCSA (Ours) | VCTK++ | 85.2 | 64.6 | 88.0 | 68.3 | 88.8 | 69.9 | 89.2 | 70.4 | 89.5 | 70.5 | 0.25 | 0.23 |
| | Libri-Dys | 79.2 | 68.5 | 79.3 | 69.7 | 79.7 | 70.2 | 80.9 | 70.2 | 81.7 | 70.9 | 0.72 | 0.21 |
| SSDM+NICL (Ours) | VCTK++ | 85.4↑ | 64.7↑ | 88.1↑ | 68.3↑ | 88.7↑ | 69.9↑ | 89.1↑ | 69.9 | 89.8↑ | 71.0↑ | 0.37 | 0.33 |
| | Libri-Dys | 79.3↑ | 68.6↑ | 79.2↑ | 69.9↑ | 79.3 | 69.9↑ | 81.9↑ | 71.9↑ | 82.8↑ | 72.4↑ | 1.31 | 0.95 |
| SSDM 2.0 (Ours) | VCTK++ | **85.7** | **65.0** | **88.5** | **68.7** | **88.9** | **70.7** | **90.4** | **71.6** | **92.6** | **73.5** | **1.26** | **2.83** |
| | Libri-Dys | **80.1** | **69.2** | **79.9** | **70.3** | **80.0** | **70.3** | **83.2** | **73.4** | **86.2** | **75.9** | **2.18** | **1.99** |

### 6.4 DISCUSSION ON THE SCALABILITY OF DYSFLUENCY DETECTION

In Section 6.3, we evaluated the scalability of our scalable representations. We subsequently assessed whether our entire system, as well as each individual module, functions as an effective and scalable dysfluency detector. As illustrated in Table 3, we systematically replaced each module in SSDM. When we substituted SSDM gestural scores with Neural Articulatory Flow (NAF), Connectionist Sequence Alignment (CSA) with Full-stack Connectionist Sequence Alignment (FCSA), and the original Language Model (LM) pipeline with our Non-fluency In-context Learning (NICL), we consistently observed substantial improvements in both detection accuracy (F1 and MS) and scaling factors (SF1 for F1 and SF2 for MS). These results indicate the effectiveness of our proposed NAF, FCSA, and NICL modules. It is noteworthy that in the original SSDM, the incorporation of LLaMA (Touvron et al., 2023) did not appear to enhance performance, as indicated by downward arrows in our results. Finally, we report our SSDM 2.0 results, which combine NAF, FCSA, and NICL. This iteration achieves state-of-the-art results, significantly outperforming SSDM. We also conducted experiments with curriculum learning (training each module separately before end-to-end training); however, we did not observe any significant performance changes with this approach.

Table 4: Scalable Dysfluent Detection Evaluation on Co-Dysfluency Corpus

| Method | Eval Data | F1 (%, ↑) | MS (%, ↑) | F1 (%, ↑) | MS (%, ↑) | F1 (%, ↑) | MS (%, ↑) | SF1 (%, ↑) | SF2 (%, ↑) |
|---|---|---|---|---|---|---|---|---|---|
| Training Data | | Libri-Dys-Co (30%) | | Libri-Dys-Co (60%) | | Libri-Dys-Co (100%) | | | |
| SSDM w/ Curri (Lian et al., 2024) | Libri-Dys-Co | 79.2 | 68.4 | 79.7 | 68.8 | 81.0 | 70.2 | 0.59 | 0.52 |
| SSDM 2.0 (Ours) | Libri-Dys-Co | **81.4** | **72.3** | **83.0** | **73.7** | **87.0** | **76.3** | **1.84** | **1.34** |

**Co-Dysfluency Detection** We conducted a comparative analysis of SSDM (Lian et al., 2024) (employing curriculum learning) and our proposed SSDM 2.0. The evaluation was performed on the Libri-Dys-Co test set, utilizing various splits of the training set. The results, presented in Table 4, demonstrate that SSDM 2.0 functions as decent and scalable co-dysfluency detector.

### 6.5 HOW MUCH CAN SLMs TACKLE (CO)DYSFLUENCY PROBLEMS?

Table 5: Results comparison to speech language models. SALMONN-13B (Tang et al., 2023), GPT4 (OpenAI et al., 2023), GPT4o (OpenAI, 2024), SSDM w/ Curri (Lian et al., 2024).

| Eval Data | SALMONN-13B | | SALMONN-13B-FT | | GPT4 | | GPT4o | | SSDM w/ Curri | | SSDM 2.0 (Ours) | |
|---|---|---|---|---|---|---|---|---|---|---|---|---|
| | F1(%, ↑) | MS(%, ↑) | F1(%, ↑) | MS(%, ↑) | F1(%, ↑) | MS(%, ↑) | F1(%, ↑) | MS(%, ↑) | F1(%, ↑) | MS(%, ↑) | F1(%, ↑) | MS(%, ↑) |
| Libri-Dys | 7.7 | 0 | 11.0 | 2.5 | 18.5 | 0 | 18.3 | 0 | 81.6 | 71.0 | **86.2** | **75.9** |
| Libri-Dys-Co | 2.4 | 0 | 13.9 | 6.8 | 15.0 | 0 | 22.9 | 0 | 81.0 | 70.2 | **87.0** | **76.3** |
| nfvPPA | 0 | 0 | 1.8 | 0 | 5.6 | 0 | 6.4 | 0 | 69.9 | 55.0 | **76.8** | **70.3** |

We compiled results from SALMONN (Tang et al., 2024), GPT4 speech API (OpenAI et al., 2023), and GPT4o real-time API (OpenAI, 2024) to evaluate performance on Libri-Dys, Libri-Dys-Co, and nfvPPA datasets. Some of these results are sourced from Lian et al. (2024). Additionally, we utilized the same data to perform instruction tuning with SALMONN (referred to as SALMONN-13B-FT). As shown in Table. 5, current Speech Language Models (SLMs) demonstrate inferior performance compared to the SSDM series in the context of dysfluency detection and transcription.

### 6.6 OTHER BENCHMARKS

Table 6: Compare with YOLO-Stutter on Different Benchmarks

| Methods | Dataset | Rep Acc.% | BL | Block Acc.% | BL | Miss Acc.% | BL | Replace Acc.% | BL | Prolong Acc.% | BL |
|---|---|---|---|---|---|---|---|---|---|---|---|
| YOLO-Stutter(VCTK-Stutter) | VCTK-Stutter Testset | 99.16 | 26ms | 99.29 | 25ms | 80.00 | 18ms | - | - | 91.84 | 35ms |
| YOLO-Stutter(VCTK-TTS) | VCTK-Stutter Testset | 83.11 | 27ms | 100 | 22ms | 40.00 | 17ms | - | - | 90.34 | 34ms |
| SSDM (VCTK-TTS) | VCTK-Stutter Testset | 100 | 25ms | 100 | 21ms | 54.60 | 16ms | - | - | 91.80 | 32ms |
| SSDM2.0 (VCTK-TTS) | VCTK-Stutter Testset | **100** | **25ms** | **100** | **21ms** | **88.50** | **15ms** | - | - | **92.00** | **32ms** |
| YOLO-Stutter(VCTK-Stutter) | VCTK-TTS Testset | 78.31 | 66ms | 92.44 | 43ms | 43.33 | 42ms | - | - | 88.17 | 42ms |
| YOLO-Stutter(VCTK-TTS) | VCTK-TTS Testset | 98.78 | 27ms | 98.71 | 78ms | 70.00 | 8ms | 73.33 | 10ms | 93.74 | 32ms |
| SSDM (VCTK-TTS) | VCTK-TTS Testset | 100 | 25ms | 100 | 66ms | 72.30 | 8ms | 74.00 | 10ms | 94.67 | 30ms |
| SSDM2.0 (VCTK-TTS) | VCTK-TTS Testset | **100** | **25ms** | **100** | **62ms** | **80.80** | **6ms** | **78.00** | **8ms** | **95.02** | **28ms** |

We also consider other decent dysfluency modeling efforts. YOLO-Stutter (Zhou et al., 2024b) adapted YOLO (Redmon, 2016), treating dysfluency detection as a time-domain object detection problem. Stutter-Solver (Zhou et al., 2024a) extends YOLO-Stutter to multilingual domain. Time-and-Tokens (Zhou et al., 2024c) revisits this problem as ASR task, discarding time-based modeling. For our comparative analysis, we focused on YOLO-Stutter and evaluated our model on their benchmark. In this context, ACC represents type accuracy, and BL denotes normalized boundary loss. The results are presented in Table 6.6, where the method is followed by the training set in the *Methods* column. Our findings demonstrate that SSDM 2.0 consistently outperforms all other methods. It is worth noting that due to the relatively small scale of VCTK-TTS and VCTK-Stutter datasets, some performance differences are not substantial, or these datasets may be considered comparatively *easy*.

### 6.7 IN-CONTEXT LEARNING: ZERO-SHOT DYSFLUENCIES AND ASR TASKS TRANSFER

To assess our Non-fluency In-Context Learning (NICL), we devised two tasks. Task-1 focuses on zero-shot dysfluency transfer: we trained the model on single dysfluency (repetition) using Libri-Dys, then evaluated it on other types (replacement, insertion, deletion). Table 7 illustrates that SSDM 2.0 exhibits significantly greater In-Context Learning capacity than SSDM. Notably, the transfer from repetition to deletion proves more challenging.

Table 7: Non-fluent In-context Learning: Zero-Shot Dyfluencies Transfer

| Method | Training Data | F1 (%, ↑) | MS (%, ↑) | F1 (%, ↑) | MS (%, ↑) | F1 (%, ↑) | MS (%, ↑) |
|---|---|---|---|---|---|---|---|
| Eval Data | | *Libri-Dys-Replace* | | *Libri-Dys-Insertion* | | *Libri-Dys-Deletion* | |
| SSDM w/ Curri | Libri-Dys-Repetition | 23.2 | 17.9 | 32.4 | 28.0 | 11.0 | 8.2 |
| SSDM 2.0 (Ours) | Libri-Dys-Repetition | **55.4** | **47.0** | **66.2** | **60.9** | **32.4** | **29.9** |
| w/o NICL | Libri-Dys-Repetition | 49.3 | 43.9 | 60.7 | 53.0 | 30.1 | 29.9 |

For Task 2, we tested zero-shot ASR capability without additional ASR training. Table 8 shows results using Whisper (Radford et al., 2023) for normal ASR, and ASR instruction for direct transcription and WER computation on the test set. While SSDM shows poor zero-shot performance, SSDM 2.0 surprisingly achieves better-than-baseline zero-shot ASR results, demonstrating its enhanced adaptability in speech recognition tasks.

Table 8: Non-fluent In-context Learning: Zero-Shot ASR tasks Transfer

| | *Libri-Dys* | *Libri-Co-Dys (Multi-types)* |
|---|---|---|
| WER (Whisper) (% ↓) | 4.167 | 8.89 |
| WER-Zero-Shot (SSDM) (% ↓) | 10.08 | 17.45 |
| WER-Zero-Shot (SSDM 2.0) (% ↓) | **3.92** | **7.10** |

## 7 CONCLUSIONS AND LIMITATIONS

We introduce SSDM 2.0, featuring Neural Articulatory Flow, Fullstack Connectionist Subsequence Aligner, and Non-fluency In-Context Learning. We open-sourced a large-scale co-dysfluency corpus *Libri-Dys-Co*. SSDM 2.0 significantly outperforms current works (Appendix. A.12). The method's potential with increased data remains unexplored. Additional future work will focus on developing fine-grained simulation techniques, addressing the primary bottleneck in this domain.

On the clinical side, due to data constraints, we evaluated only nfvPPA for articulation-based dysfluencies, leaving out other disorders such as Parkinson's disease and Broca's aphasia. It would also be valuable to extend this work to semantic-based dysfluency disorders, such as svPPA, Wernicke's aphasia, and ASD, to enhance the pipeline's applicability as a general speech transcription tool for a broader range of speech disorders. This will be addressed in future work.

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

# A APPENDIX

## A.1 DYSFLUENCY SIMULATION

### A.1.1 METHOD

We utilize the TTS-based method (Zhou et al., 2024b) to perform dysfluency simulation. To scale our *Libri-Co-Dys* using the LibriTTS (Zen et al., 2019) corpus, we choose StyleTTS2 (Li et al., 2023) as our TTS synthesizer. For **Single-Type** Co-dysfluency, we insert 2-3 instances of the same type of dysfluency (TTS rules for each type of dysfluency are detailed in Zhou et al. (2024b)) at various positions within an utterance. For **Multi-type** Co-dysfluency, we incorporate 5 combinations of dysfluencies: (rep-missing), (rep-block), (missing-block), (replace-block) and (prolong-block), with 2 random positions chosen for each combination within the utterance. Fig. 6 shows the distribution of various types of dysfluency in the *Libri-Co-Dys* corpus. The pipeline of simulation are detailed in Fig. 5. We have open sourced *Libri-Co-Dys* at https://bit.ly/3Y5boyZ.

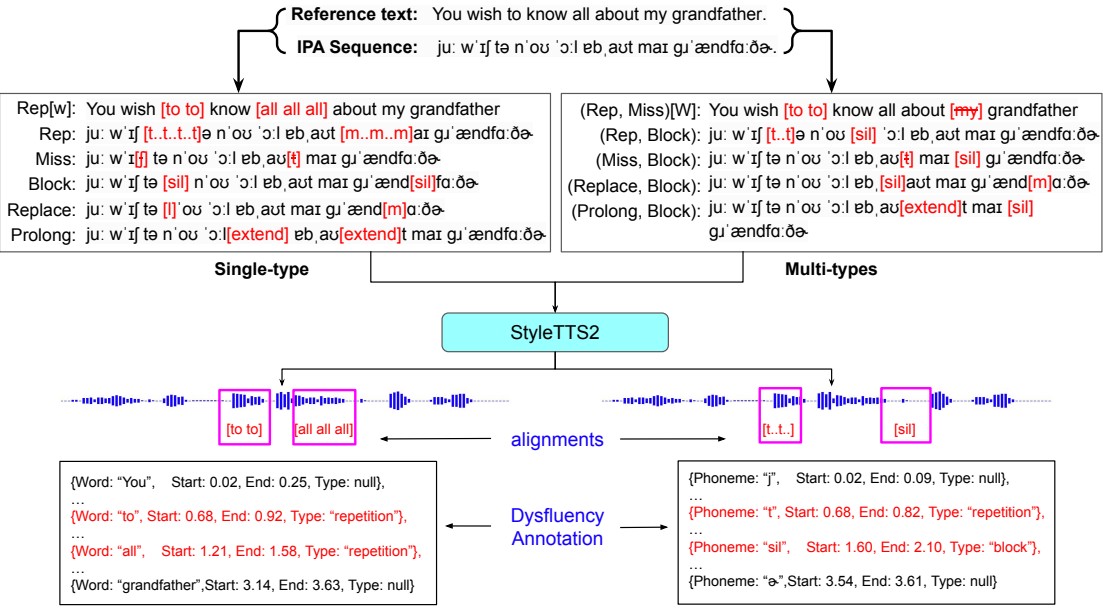

Figure 5: Dysfluency Simulation Pipeline: We first convert reference text of LibriTTS into IPA sequences via the phonemizer (Bernard & Titeux, 2021), then inject different types and groups of dysfluencies according to the TTS rules (Zhou et al., 2024b).We take dysfluency-injected IPA sequences as inputs, conduct the StyleTTS2 (Li et al., 2023) inference procedure and obtain the dysfluent speech. Finally We retrieve alignments from StyleTTS2 duration model, annotate the type of dysfluency on the dysfluent region.

We also visualize **Soft speech-text alignment** in Appendix. A.13 to highlight the challenges in dysfluency simulation and detection.

### A.1.2 SIMULATED DATASETS

- **VCTK++ (Lian et al., 2023b)** For each waveform in the VCTK (Yamagishi et al., 2019) corpus, dysfluencies such as repetitions, prolongations, and blocks were simulated by directly injecting them into the acoustic space, using forced alignments from the Montreal Forced Aligner (MFA) (McAuliffe et al., 2017).
- **VCTK-Stutter (Zhou et al., 2024b)** extends VCTK++ by incorporating word-level repetitions and deletions. Similar to how phoneme-level alignments are obtained using the MFA, VCTK-Stutter employs WhisperX (Bain et al., 2023) to acquire word-level alignments. The word-level dysfluencies are also injected at the acoustic level.

- **VCTK-TTS (Zhou et al., 2024a)** is a TTS-based simulated dataset extended from VCTK. Dysfluencies at both phoneme and word level including repetition, missing, block, replacement and prolongation are injected into the text space, and a text-to-speech model - VITS (Kim et al., 2021) - is used to generate the dysfluent speech and corresponding alignment.

Fig. 7 compares the number of types and scale of currently available simulated datasets. *Libri-Co-Dys* demonstrates significant advantages in both type diversity and dataset size.

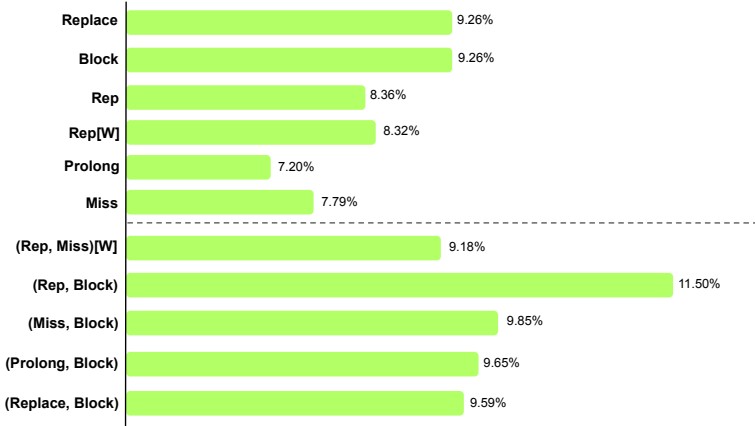

Figure 6: Distribution of Dysfluency types in Libri-Co-Dys

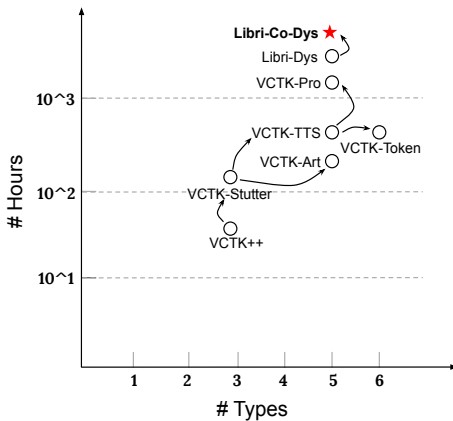

Figure 7: Comparison of Existing Simulated Dysfluency Datasets: The arrows represent the chronological order and logical relationships in the creation of the dataset."

## A.2 NFVPPA

In our work, we choose to concentrate on a specific neurodegenerative disease named nonfluent variant primary progressive aphasia (nfvPPA) for testing our pipeline. This phenotype is one of the three distinct forms of primary progressive aphasia (PPA), a group of disorders characterized by initially having most prominent disturbances to speech and language capabilities. The variants of PPA - semantic (svPPA), logopenic (lvPPA), and nonfluent (nfvPPA) (Gorno-Tempini et al., 2011) - each display unique clinical symptoms and distinct patterns of brain degeneration. Disturbances to speech fluency can occur due to multiple underlying causes subsuming different speech and language subsystems in all of these variants; among these, nfvPPA is particularly noted for its impact on speech dysfluency, characterized by primary deficits in syntax, motor speech (i.e., in this case, apraxia of speech), or both. Its association with apraxia of speech makes nfvPPA an ideal candidate for assessing automatic processing of dysfluent speech.

Our collaborators are engaged in an observational research study where they recruit patients diagnosed with this disease to participate in detailed speech and language assessments conducted by a qualified speech-language pathologist (SLP). These assessments includes a thorough motor speech evaluation, which includes an oral mechanism exam, diadochokinetic rates, maximum phonation time, reading multisyllabic words, words of increasing length, reading passages, and connected speech samples. For our present purposes, we are focusing on the speech reading of participants as they read aloud the Grandfather Passage, a passage frequently used clinically to assess motor speech due to its inclusion of nearly all phonemes of the English language. We have recordings for 38 participants with nfvPPA, captured using high-quality microphones during both in-person and remote sessions. Note that nfvPPA data will not be released.

### A.2.1 SEGMENTATION AND ANNOTATION

We first utilize the denoiser (Defossez et al., 2020) on all recordings. Subsequently, each recording was manually segmented into 15 (or less) clips, the segmentation rule of grandfather passage is as follows:

> *You wish to know all about my grandfather*
> *Well, he is nearly 93 years old*
> *yet he still thinks as swiftly as ever*
> *He dresses himself in an old black frock coat*
> *usually several buttons missing*
> *A long beard clings to his chin*
> *giving those who observe him a pronounced feeling of the utmost respect*
> *When he speaks*
> *his voice is just a bit cracked and quivers a bit*
> *Twice each day he plays skillfully and with zest upon a small organ*
> *Except in the winter when the snow or ice prevents*
> *he slowly takes a short walk in the open air each day*
> *We have often urged him to walk more and smoke less*
> *but he always answers, "Banana oil!"*
> *Grandfather likes to be modern in his language*

We have developed a complete nfvPPA annotation pipeline, which is detailed in Fig. 8.

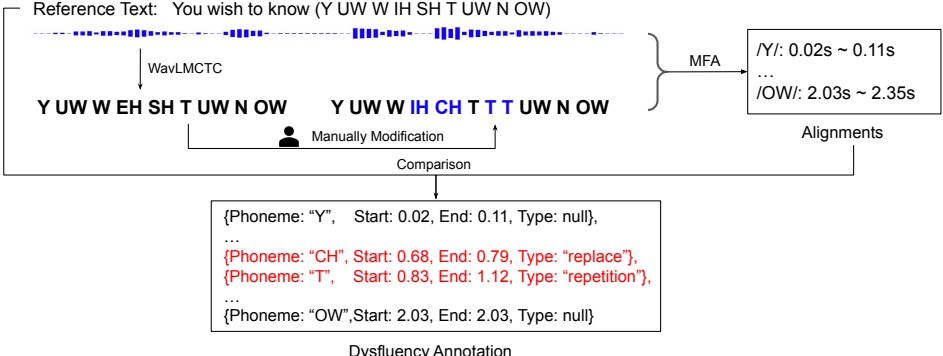

Figure 8: nfvPPA Annotation Pipeline: We first acquire the initial CMU phoneme transcriptions from the denoised audio recordings using the WavLM-CTC (Microsoft, 2021). These transcriptions are subsequently manually modified to enhance its accuracy. Following this, the refined transcriptions are processed through the Montreal Forced Aligner (MFA) (McAuliffe et al., 2017) to obtain precise phoneme alignments. We then perform a comparison between the reference text and the phoneme alignments, obtain the annotations of dysfluencies, which are incorporated as key "Type" in a JSON file.

### A.3 EVALUATION

#### A.3.1 PHONETIC TRANSCRIPTION(ALIGNMENT) EVALUATION

To assess the precision of phoneme recognition transcription at the frame level, we take the **F1 Score** (Lian et al., 2023b) as evaluation metric. F1 score measures how many phonemes are correctly predicted, which is different from Strgar & Harwath (2023) that focuses on the accuracy of predicting phonetic boundaries in terms of time steps. Additionally, to evaluate the performance of phoneme segmentation performance in our methods, we utilize the duration-aware phoneme error rate (**dPER**) (Lian et al., 2023b). dPER extends traditional Phoneme Error Rate (PER) by assigning weights to each type of error - substitution, insertion, and deletion - based on their duration. Denote $\hat{S}, \hat{I}, \hat{D}, \hat{C}$ as the weighted value of substitutions, insertions, deletions, and correct samples respectively. We compare phoneme $p_i$ and $p_j$ from the reference and predicted sequences, with $d(p_i)$ and $d(p_j)$ representing their respective durations. The update rule for each detected error type is proposed following: $\hat{S} \to \hat{S}+d(p_i)+d(p_j), \hat{I} \to \hat{I}+d(p_j), \hat{D} \to \hat{D}+d(p_i), \hat{C} \to \hat{C}+|d(p_i)-d(p_j)|$. The ultimate formula is:

$$\text{dPER} = \frac{\hat{S}+\hat{D}+\hat{I}}{\hat{S}+\hat{D}+\hat{C}} \tag{12}$$

#### A.3.2 DYSFLUENCY EVALUATION

We evaluate dysfluency in segments of Aphasia speech through annotations that capture all types of dysfluencies and corresponding accurate timings. We assess the identification of dysfluency types using **F1 Score**. Additionally, the accuracy of dysfluency detection in terms of time alignment is measured by calculating the Intersection over Union (IoU) between the predicted time and the ground truth time boundaries. A dysfluency is considered accurately detected if the IoU exceeds 0.5. We also compute an F1 score for this matching evaluation, referred to as the **Matching Score (MS)**. The illustration of these metrics is shown in Fig. 9.

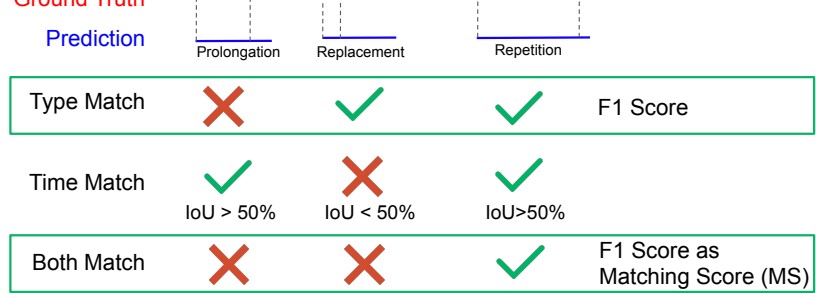

Figure 9: Metrics of Dysfluency Evaluation

### A.4 NEURAL IMPLICIT SPEECH REPRESENTATIONS

Current speech representation modeling typically uses explicit $T \times D$ matrices, where T is time and D is channel dimension. However, human speech is produced by a limited set of articulators with sparse activation in time (Browman & Goldstein, 1992), forming structured sparse representations (Ramanarayanan et al., 2013) called *gestural scores*. This sparse representation concept has been applied in fields like face recognition (Wright et al., 2008). When a feature's physical structure is known, implicit representations can be employed, as explored in Mildenhall et al. (2021). Can we develop functions for implicit speech representations (*gestural scores*) as alternatives to explicit dense matrices like mel-spectrograms or self-supervised units (Mohamed et al., 2022). Implicit representations offer greater efficiency and scalability due to their sparse nature. Previous work has explored deriving spatial and temporal sparse activation matrices via matrix factorization (Lian et al., 2022; 2023a) or complex entry-wise joint-duration-intensity modeling (Lian et al., 2024). We propose to derive implicit *gestural scores*.

## A.5 POST-INTERPRETABLE TRAINING

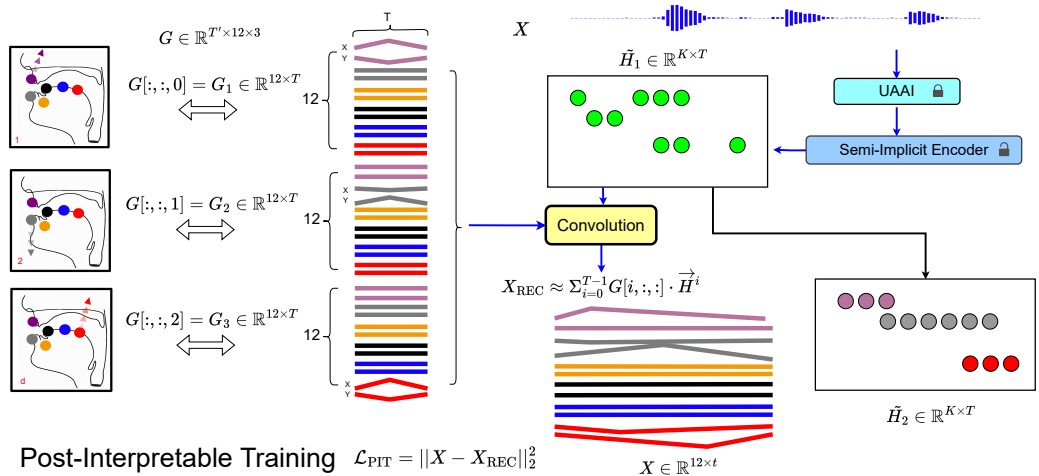

Figure 10: Illustration of Articulatory Gestures and Post-Interpretable Training

Articulatory data $X \in \mathbb{R}^{D \times T}$ essentially represents a sequence of motion data. The state-of-the-art acoustic-to-articulatory inversion (AAI) method (Cho et al., 2024) has demonstrated fully intelligible speech synthesis performance, thus it can be considered a powerful *articulatory-free* representation. The term *articulatory-free* signifies that actual bio-signal data is not required, and the articulatory trajectory from AAI is analogous to other speech features such as mel-spectrograms or self-supervised units (Mohamed et al., 2022). Consequently, speech can also be conceptualized as motion data. Any motion data can be decomposed into a set of bases (primitives) of moving patterns and their activations. In robotics, this concept is referred to as a gait library (Grizzle et al., 2010), while in speech, it is termed gestures (cases) and gestural scores (activations) (Browman & Goldstein, 1992). We provide a simple example to illustrate this concept and its computation. As shown in Fig. 10, we have gestures $G \in \mathbb{R}^{T' \times 12 \times K}$ where $T'$ is the window size, 12 represents the x, y coordinates of 6 articulators, and K denotes the number of gestures. It should be noted that K=3 is used here for visualization purposes only. In the actual implementation, 40 kernels are utilized, matching the size of the CMU dictionary. For post-interpretable training, given semi-implicit gestural scores $\hat{H}_1 \in \mathbb{R}^{K \times T}$, we perform 1D convolution with these gestures as convolution kernels. This process reconstructs $X_{\text{REC}} \approx \Sigma_{i=0}^{T-1} G[i,:,:] \cdot \overrightarrow{H}^i$. The reconstruction loss is defined as $\mathcal{L}_{\text{PIT}} = ||X - X_{\text{REC}}||_2^2$. Following post-interpretable training, we obtain interpretable gestural scores $\hat{H}_2$, which provide precise information about articulatory movements and their correspondence to speech production. For instance, in $\hat{H}_2$, we observe a sequence of upper lip elevation, lower lip elevation, and finally, tongue dorsum elevation. To elaborate:

- Upper lip elevation suggests a bilabial constriction (bringing both lips together), typically associated with sounds like /p/ or /b/.

- Lower lip elevation, when the upper and lower lips are already in proximity, reinforces a bilabial closure, further supporting the likelihood of a /p/ or /b/ sound.

- Tongue dorsum elevation involves raising the back of the tongue, characteristic of velar sounds such as /k/ or /g/.

The combination of these articulatory movements most likely generates a sound sequence like /p/ or /b/ followed by /k/ or /g/. This articulatory sequence is commonly associated with consonant clusters found in various languages. In English, for example, a similar sequence occurs in words such as "back" (/bæk/) or "pack" (/pæk/), where a bilabial sound (/p/ or /b/) precedes a velar (/k/ or /g/) sound.

*The primary objective of conducting post-interpretable training is to visualize the origins of mispronunciations. By applying gradient-weighted class activation mapping (Grad-CAM) (Selvaraju et al., 2017) to visualize the gradient of the interpretable gestural scores $\hat{H}_2$, it becomes feasible to precisely locate articulatory issues. This approach facilitates the provision of articulatory-aware feedback, as proposed by Lian et al. (2024).*

## A.6 DISCUSSION ABOUT INTERPRETABLE ARTICULATORY FEEDBACK

In Section 3.3, we introduced Post-Interpretable Training to enhance gestural score interpretability, enabling pronunciation error localization via gradCAM (Selvaraju et al., 2017) (Appendix A.5). Originally proposed in Lian et al. (2024), it lacks objective evaluation metrics. We employ it as an optional pronunciation assistance tool without formal evaluations which are left for future work.

## A.7 LOCAL SEQUENCE ALIGNMENT

For fluent speech, this alignment is strictly monotonic. A common approach involves identifying local non-monotonic alignments by excluding monotonic segments (Lian et al., 2023b). The latest methodology (Lian et al., 2024) maintains a monotonic alignment paradigm even when addressing speech disfluencies. For example, given the reference text *P-L-IY-Z* and the spoken sequence *P-P-L-EY-SIL-EY-Z*, the alignment is: *[P-[P,P], L-[L], IY-[EY,SIL-EY], Z-[Z]]*. In this structure, the ground truth text is followed by its corresponding speech elements, highlighting phenomena such as stuttering ("P"), blocking and phonetic errors ("IY" vs. "EY"), while other pronunciations match the reference text. Lian et al. (2023b) posited that such an alignment *[P-[P,P], L-[L], IY-[EY,SIL-EY], Z-[Z]]* can be derived via the longest common subsequence (LCS) algorithm (Hirschberg, 1977). LCS is a local sequence alignment algorithm; by *local*, it means that the cost function only considers entries where a speech frame matches a text token while disregarding other tokens, which is crucial for capturing disfluencies. This approach differs significantly from global sequence aligners such as Dynamic Time Warping (DTW) (Sakoe, 1971), where all entries contribute to the cost function and thus are not well-suited for modeling non-fluent speech. However, speech tokens are more abstract than phonemes. Directly applying LCS does not necessarily yield a dysfluency-aware alignment, and there could be multiple reasonable alignments given speech sequence and text sequence. Consequently, a differentiable, stochastic subsequence aligner is required.

SSDM (Lian et al., 2024) introduced connectionist subsequence alignment (CSA) as the first proper estimation method. However, there are notable limitations. (1) SSDM primarily focuses on *Transition Skip* $y^{i-k,j-1} \rightarrow y^{i,j}$ for $k > 1$, capturing dysfluencies like repetition, blocking, and insertion, while other transition types, such as word/phoneme omission $y^{i-k,j-k} \rightarrow y^{i,j}$, are not explicitly addressed. Additionally, the emission probability $y^{i,j}$ can be *passive* or skipped, which SSDM overlooks. (2) The adapted forward-backward algorithm lacks interpretability regarding the specific dysfluency patterns encoded. (3) The neural gestural score $H$ is trained using a separate phoneme classification task, adding to training complexity. In this work, we address the aforementioned problems by introducing *Pre-Alignment Training*, which incorporates full-stack transition modeling with clear interpretability and controllability.

## A.8 INTERPRETE CONNECTIONIST SUBSEQUENCE ALIGNER

Strictly speaking, the original CSA (Lian et al., 2024) explicitly encodes stack-1 and partially encodes stack-3 to some extent (albeit with a potentially improper decay). The emission copy ($y^{i-1,j} \rightarrow y^{i,j}$, primarily encoded by $\alpha^{i-1,j} \rightarrow \alpha^{i,j}$) implicitly encodes stack-2. However, this approach presents several significant limitations:

- The weights for each stack ($\delta_k$) are predefined, lacking flexibility for adjustment.
- Stack-3 is only partially encoded, and some transitions lack logical consistency, potentially introducing noise.
- Stacks 4-6 are entirely omitted from the encoding process.

These factors constitute significant limitations of the vanilla CSA. To elucidate these points, we can decompose the original formula presented in Lian et al. (2024). We examine the forward algorithm

in Eq. 13 and backward algorithm in Eq. 15.

$$\alpha^{i,j} = \alpha^{i-1,j} + \sum_{k=1}^{j} \delta^k \alpha^{i-1,j-k} \cdot y^{i,j} \cdot \left( p_\theta(C_{j-1}^S | C_j^S) \cdot \mathbf{1}_{\{k=1\}} + \mathbf{1}_{\{k \neq 1\}} \right)$$

$$= \alpha^{i-1,j} + \delta \alpha^{i-1,j-1} \cdot y^{i,j} \cdot \left( p_\theta(C_{j-1}^S | C_j^S) \right) \text{ (Stack-1)} \tag{13}$$

$$+ \sum_{k=2}^{j} \delta^k \alpha^{i-1,j-k} \cdot y^{i,j} \text{ (Partial Stack-3)} \tag{14}$$

$$\beta^{i,j} = \beta^{i+1,j} + \sum_{k=1}^{T-j} \delta^k \beta^{i+1,j+k} \cdot y^{i,j} \cdot \left( p_\theta(C_j^S | C_{j+1}^S) \cdot \mathbf{1}_{\{k=1\}} + \mathbf{1}_{\{k \neq 1\}} \right)$$

$$= \beta^{i+1,j} + \delta \beta^{i+1,j-1} \cdot y^{i,j} \cdot \left( p_\theta(C_j^S | C_{j+1}^S) \right) \text{ (Stack-1)} \tag{15}$$

$$+ \sum_{k=2}^{T-j} \delta^k \beta^{i+1,j+k} \cdot y^{i,j} \text{ (Partial Stack-3)} \tag{16}$$

## A.9 NEURAL BACKWARD PROCESS

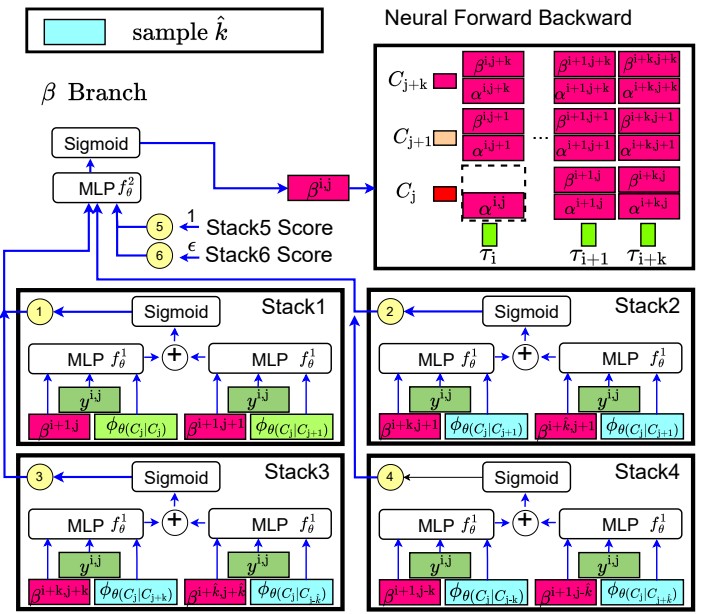

Figure 11: Backward Process ($\beta$ Branch)

Due to page limit constraints, we only list the forward process in the main text. However, we also have a backward process, as shown in Eq.17 and Eq.18.

$$\beta_1^{i,j} = f^0 \left( f_\theta^1(\beta^{(i+1,j)}, \phi_\theta(C_j | C_j), y^{i,j}) + f_\theta^1(\beta^{(i+1,j+1)}, \phi_\theta(C_j | C_{j+1}), y^{i,j}) \right) \tag{17}$$

Stacks 2-4 correspond to the non-fluency forward process, which is uniformly shown in Eq. 18.

$$\beta_u^{i,j} = f^0 \left( f_\theta^1(\beta^{(i+a_u,j+b_u)}, \phi_\theta(C_j | C_{j+b_u}), y^{i,j}) + f_\theta^1(\beta^{(i+\hat{a}_u,j+\hat{b}_u)}, \phi_\theta(C_j | C_{j+\hat{b}_u}), y^{i,j}) \right) \tag{18}$$

where $u \in \{2,3,4\}, (a_2, b_2) = (k,1), (\hat{a}_2, \hat{b}_2) = (\hat{k}, 1), (a_3, b_3) = (k, k), (\hat{a}_3, \hat{b}_3) = (\hat{k}, \hat{k}), (a_4, b_4) = (1, -k), (\hat{a}_4, \hat{b}_4) = (1, -\hat{k})$, and $k \leq \hat{k} \leq \min(\max(i) - i, \max(j) - j) - 1$ are randomly sampled to increase dysfluency diversity.

## A.10 SAMPLING PROCESS

Following the completion of both *Pre-Alignment Training* and *Post-Alignment Training*, we obtain speech representations $\tau = [\tau_1, \tau_2, \ldots, \tau_T]$, text tokens $C = [C_1, C_2, \ldots, C_L]$, and the transition probability function $\phi_\theta(\cdot|\cdot)$. Additionally, we have the emission probability: $y^{i,j} = p_\theta(C_j|\tau_i) \approx \frac{\exp(\tau_i \cdot C_j^S)}{\sum_{k=1}^{L} \exp(\tau_i \cdot C_k^S)}$ where $C_j^S$ is sampled from the normal distribution $\mathcal{N}(\mu_\theta^{C_j}, (\sigma_\theta^{C_j})^2)$. These elements collectively define the distribution of all non-fluency alignments $\Gamma(C, \tau)$. To sample an alignment from this distribution, we employ the longest common subsequence algorithm (Hirschberg, 1977), which has demonstrated superior performance compared to traditional search algorithms such as beam search. Our proposed algorithm is delineated as follows:

## A.11 LANGAUGE MODELING

For model setup and configurations, we follow Gong et al. (2023b); Lian et al. (2024) regarding text encoder and embedding sizes (4096). For the LoRA module, we set the rank to 8 and $\alpha = 16$. The non-fluent speech text alignment $\gamma(C, \tau)$, sampled from the longest common subsequence algorithm, is concatenated frame-wise. Details are as follows: Let $\gamma^{-1}(\tau_i)$ denote the text aligned to speech token $\tau_i$, where $\gamma^{-1}$ is the inverse function of $\gamma$. Given speech sequences $\tau = [\tau_1, \tau_2, \ldots, \tau_T] \in \mathbb{R}^{D \times T}$, we obtain text tokens aligned to speech tokens as $C = [\gamma^{-1}(\tau_1), \gamma^{-1}(\tau_2), \ldots, \gamma^{-1}(\tau_T)] \in \mathbb{R}^{D \times T}$. We concatenate at each time frame to obtain a $2D \times T$ matrix, followed by one MLP ($2D \times 4096$) to form the final inputs, where $D = 64$. For time modeling, we follow Huang et al. (2024), converting our time annotations (Appendix A.1) to frame indices for prediction. We use a final *interface* prompt to convert predicted frame indices back and refine the output:

> *Please return the output via the following format: The speaker is attempting to speak the ground truth text <1>. We are going to analyze the pronunciation problem for each word:*
>
> - *For word <1>, the pronunciation problems are <2> at time <3>.*
> $\vdots$
> - *For the last word <1>, the pronunciation problems are <2> at time <3>.*
>
> *[End of Template] Instructions for filling the template:*
>
> 1. *Replace <1> with actual words.*
> 2. *Replace <2> with actual non-fluencies.*
> 3. *Replace <3> with either a time step or time range.*
>     - *If the time range is too short (< 0.1s), only return the start time for visualization.*
>     - *Convert frame-indices to exact time, considering each frame is 0.02s.*
>
> *Note: You may adjust the text for flexibility as needed, without strictly adhering to this template structure.*

We also have a prompt to only extract dysfluency type and time information for evaluation, such as the computation of F1 score and MS score. The prompt is listed in the following:

> *Please return the output in a JSON-friendly format, which includes the following fields:*
>
> - `word`: *the word being analyzed*
> - `dysfluency`: *the identified pronunciation problem (e.g., repetition, prolongation)*
> - `time_start`: *the start time (in seconds)*
> - `time_end`: *the end time (in seconds, if applicable, otherwise leave it null)*
>
> *The format for each word should be as follows:*
>
> ```
> \{
>   "word": "<word>",
> ```

---

**Algorithm 1** Sampling Alignment $\gamma(C, \tau)$ during both Training and Inference

---

1: **Input:** Speech representations $\tau = [\tau_1, \tau_2, \ldots, \tau_T]$
2: **Input:** Text tokens $C = [C_1, C_2, \ldots, C_L]$
3: **Output:** Alignment $\gamma(C, \tau) = [(\tau_1, C_{\text{aligned to } \tau_1}), \ldots, (\tau_T, C_{\text{aligned to } \tau_T})]$
4: Initialize $dp$ table of size $(T + 1) \times (L + 1)$ with all zeros
5: Initialize $\gamma(C, \tau)$ array of length $T$ with $None$
6: **for** $i = 1$ to $T$ **do**
7:     **for** $j = 1$ to $L$ **do**
8:         Compute emission probability: $emission\_prob = p_\theta(C_j | \tau_i)$
9:         **if** $j > 1$ **then**
10:             Compute transition probability: $transition\_prob = \phi_\theta(C_j | C_{j-1})$
11:         **else**
12:             $transition\_prob = 1$            ▷ No transition for the first token
13:         **end if**
14:         $combined\_prob = emission\_prob \times transition\_prob$
15:         **if** $combined\_prob > threshold$ **then**
16:             $dp[i][j] = dp[i-1][j-1] + 1$       ▷ Match: move diagonally in DP table
17:         **else**
18:             $dp[i][j] = \max(dp[i-1][j], dp[i][j-1])$    ▷ No match: take max of top or left
19:         **end if**
20:     **end for**
21: **end for**
22: Backtrack to find $\gamma(C, \tau)$:
23: $i = T, j = L$
24: **while** $i > 0$ and $j > 0$ **do**
25:     Compute emission probability: $emission\_prob = p_\theta(C_j | \tau_i)$
26:     **if** $j > 1$ **then**
27:         Compute transition probability: $transition\_prob = \phi_\theta(C_j | C_{j-1})$
28:     **else**
29:         $transition\_prob = 1$
30:     **end if**
31:     $combined\_prob = emission\_prob \times transition\_prob$
32:     **if** $combined\_prob > threshold$ **then**
33:         $\gamma(C, \tau)[i-1] = (\tau_i, C_j)$          ▷ Store alignment of $\tau_i$ with $C_j$
34:         $i = i - 1$
35:         $j = j - 1$
36:     **else if** $dp[i-1][j] > dp[i][j-1]$ **then**
37:         $i = i - 1$
38:     **else**
39:         $j = j - 1$
40:     **end if**
41: **end while**
42: **for** $i = 1$ to $T$ **do**
43:     **if** $\gamma(C, \tau)[i] = None$ **then**
44:         $\gamma(C, \tau)[i] = (\tau_i, None)$          ▷ No alignment found for $\tau_i$
45:     **end if**
46: **end for**
47: **Return** $\gamma(C, \tau)$

---

```
        "dysfluency": "<dysfluency_type>",
        "time_start": <start_time_in_seconds>,
        "time_end": <end_time_in_seconds_or_null>
    \}
```

*The final JSON object should be an array of entries, where each entry corresponds to a word, its dysfluency, and the respective time information.*

**Instructions for filling this format:**

1. Replace `<word>` with the actual word from the ground truth text.
2. Replace `<dysfluency_type>` with the specific non-fluency issue encountered.
3. Replace `<start_time_in_seconds>` with the exact time (in seconds) corresponding to the start of the non-fluency event.
4. If applicable, replace `<end_time_in_seconds_or_null>` with the time when the event ends. If the time range is very short ($< 0.1$s), only provide the start time and set `time_end` as `null`.

*For example:*

```
[
  \{
    "word": "Hello",
    "dysfluency": "prolongation",
    "time_start": 0.50,
    "time_end": 0.70
  \},
  \{
    "word": "world",
    "dysfluency": "repetition",
    "time_start": 1.20,
    "time_end": null
  \}
]
```

*Notes:*

- Ensure that frame indices are converted to seconds (with 1 frame = 0.02s).
- If a dysfluency spans over a range of time, include both `time_start` and `time_end`. Otherwise, only provide the `time_start` and set `time_end` as `null`.

For $\mathcal{L}_{\text{FINAL}}$, we set $\lambda_1 = \lambda_2 = \lambda_3 = \lambda_4 = \lambda_5 = \lambda_6 = 1$.

### A.12 MODEL ARCHITECTURE AND HYPERPARAMETERS

**Acoustic Encoder** We employ WavLM (Chen et al., 2022) large (50Hz, dimension=768) for $\hat{X}$.

**UAAI** A pretrained acoustic-to-articulatory inversion model (Cho et al., 2024) generates 50Hz, 12-dimensional representations $X$.

**Count Encoder** We utilize a one-layer MLP (12,512), projected to dimension 512, followed by a 3-layer Transformer (Vaswani et al., 2017) Base. This is succeeded by time-pooling ($768 \times T \to D \times 1$) and another 1D convolutional module (1x1 kernel is applied to predict $q_\theta(Z_C|X)$).

**Index Generator** This component employs an identical architecture to the Count Encoder, generating indices multiple times based on the Count Encoder's output.

**Index Compiler** This module extracts the corresponding column vectors for input to the Value Encoder.

**Value Encoder**  The Value Encoder processes a $T'' \times 12$ tensor, outputting $T'' \times 2$ values with both means and variances. It consists of a three-layer Transformer Base (512), with an initial MLP (12,512) and a final MLP (512,2).

**Interpretable Posterior Training**  We implement the same one-layer convolution decoder as Lian et al. (2022), with a kernel size matching the gesture sizes $T' \times 12 \times 40$, where $T'$ (window size) is 200ms.

**Articulatory Flow**  We largely adhere to the Voicebox (Le et al., 2024) flow matching configuration. We set $\sigma_{min} = 0.01$, and $v_t$ is a 6-layer Transformer encoder (dimension=512). This is preceded by an MLP ((K+2D),512)=(64,512) and followed by another MLP (512,768) to predict WavLM features.

**FCSA**  The sole learnable module is the transition probability $\phi_\theta(C_i|C_j)$, implemented as a (64, 64) linear layer with sigmoid activation, following SSDM (Lian et al., 2024). We adopt the same text (phoneme encoder) as Ren et al. (2020).

**Language Modeling**  In accordance with Lian et al. (2024), we employ the same text encoder as Gong et al. (2023a). All embedding sizes are 4090 (Touvron et al., 2023; Gong et al., 2023a). Following Gong et al. (2023a), we use a rank of 8 and $\alpha = 16$ in LoRA (Hu et al., 2021). All other settings remain constant.

**Training Settings**  In Equation 1, $\tau = 2$. We utilize Adam (Kingma & Ba, 2014) with a learning rate decay from 0.001 at a rate of 0.9 every 10n steps, consistent with Lian et al. (2024). Our model is trained on two A6000 GPUs. Notably, while SSDM (Lian et al., 2024) requires approximately 3000 steps to converge, SSDM 2.0 achieves convergence in only 285 steps.

## A.13 SOFT SPEECH-TEXT ALIGNMENT

Soft speech-text alignment is an intermediate product when simulating dysfluent speech. We obtain $|c_{text}| \times |z|$ monotonic attention matrix $A$ from StyleTTS2 (Li et al., 2023)'s duration model, which indicates how each input phoneme aligns with target speech, where $c_{text}$ is text dimension and $z$ the speech duration (with the horizontal axis denoting speech and the vertical axis denoting text). From the graph, we can observe non-monotonic and various noisy, jumping phonemes, as monotonicity is severely disrupted. This disruption poses significant challenges for dysfluency simulation and detection for future work.

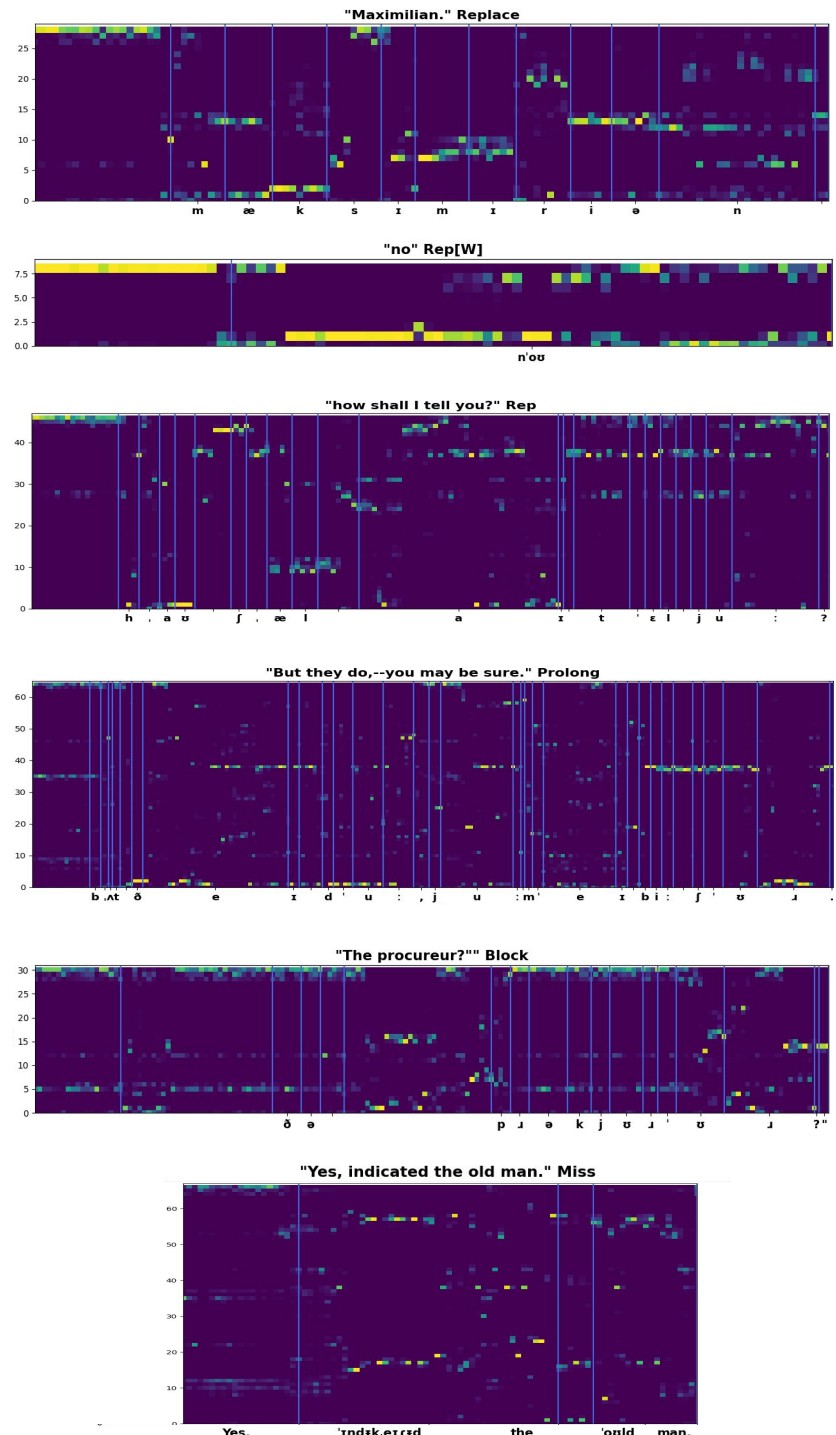

## A.14 EFFICIENCY DISCUSSION

Due to space constraints, we did not elaborate extensively on this topic in the main text. We will now discuss the efficiency of our proposed methods from two perspectives:

(1) As detailed in Appendix A.12, SSDM 2.0 demonstrates a tenfold improvement in training complexity or convergence rate compared to SSDM.

(2) In SSDM, the neural gestural scores are represented by a $K \times T$ matrix, where $K = 40$ and $T$ ranges from 100 to 2000. In contrast, SSDM 2.0 employs a sparse matrix representation for gestural scores. Our experimental observations, corroborated by Lian et al. (2022), indicate that typically only a maximum of 10% of the entries are non-zero. Given our utilization of PyTorch sparse matrix operations, our Neural Articulatory Flow (NAF) representation is demonstrably more efficient than the gestural scores in SSDM.

## A.15 DYSFLUENCY TRANSCRIPTION AND ASR ON FLUENT CORPUS

In this ablation study, we test our model's capacity on fluent speech. We focus on two tasks: dysfluency detection (transcribing what the person actually said) and ASR (transcribing what the person intended to say). Since fluent speech is assumed to have no dysfluencies, we only report false positives (FP), ignoring time information. For FP computation, we only consider the binary presence or absence of dysfluencies. For each sample, we designed an additional prompt to generate a binary score (0 or 1), where 1 indicates the presence of dysfluencies. The FP rate is computed as the sum of "1"s divided by the total number of samples. The prompt is as follows:

*Please analyze the transcript and determine whether there is any dysfluency based on the existence of entries in the following format:*
```
{ "word":  "<word>",
"dysfluency":  "<dysfluency_type>",
"time_start":  <start_time_in_seconds>,
"time_end":  <end_time_in_seconds_or_null>
}
```
*If there is at least one entry matching this format, return the following JSON object indicating dysfluency exists:*
```
{ "has_dysfluency":  1
}
```
*If no such entry exists, return the following JSON object indicating no dysfluency:*
```
{ "has_dysfluency":  0
}
```
**Instructions for evaluation:**

1. Process the transcript and identify dysfluencies.

2. If any dysfluency is found, create at least one JSON entry in the specified format.

3. Use the presence or absence of these entries to determine the value of `has_dysfluency`:
   - Set `has_dysfluency` to `1` if there is at least one entry.
   - Set `has_dysfluency` to `0` if no entry is generated.

**Example Outputs:**

1. **Transcript with Dysfluency:**
```
[
  {
    "word": "I",
    "dysfluency": "repetition",
    "time_start": 0.50,
    "time_end": null
  }
]
```

```
    {
       "has_dysfluency": 1
    }
```

2. **Transcript without Dysfluency:**

```
    []
    {
       "has_dysfluency": 0
    }
```

*Notes:*

- Ensure that any detected dysfluency generates a properly formatted JSON entry.
- The decision for `has_dysfluency` depends solely on the presence or absence of such entries.
- Maintain compatibility with Overleaf by ensuring proper escaping and formatting.

For ASR tasks, we use Whisper V2 Radford et al. (2023) as a baseline. We also use GPT4-o real-time speech API (OpenAI, 2024) to test the false positives. We first test performance in a zero-shot setting on the LibriTTS corpus. We then fine-tune our model on LibriTTS, where the target dysfluency labels are set to "None," denoted as SSDM 2.0-Tuned, with results shown in Table 9. We can see that Whisper delivers the best ASR results due to its scaling efforts. GPT4-o speech real-time interface produces some false positives on fluent speech. SSDM (Lian et al., 2024) has worse FP and WER scores. SSDM 2.0 has better FP than GPT4-o but worse ASR performance than Whisper. After "fluent" fine-tuning, SSDM 2.0-Tuned achieves the best FP scores and ASR performance comparable to Whisper.

Table 9: Evaluation on Fluent Speech

| Eval Data | LibriTTS-Test-Clean | | LibriTTS-Test-Other | |
|---|---|---|---|---|
| | FP (%, ↓) | WER (%, ↓) | FP (%, ↓) | WER (%, ↓) |
| Ground Truth | 0 | - | 0 | - |
| Whisper (Radford et al., 2023) | - | **2.7** | - | **6.3** |
| GPT4-o (OpenAI, 2024) | 14.3 | - | 14.7 | - |
| SSDM w/ Curri (Lian et al., 2024) | 37.4 | 16.5 | 39.3 | 19.9 |
| SSDM 2.0 (Ours) | **13.4** | 4.3 | **13.7** | 7.6 |
| SSDM 2.0-Tuned (Ours) | **7.4** | 3.3 | **9.2** | 6.6 |

## A.16 DYSFLUENCY TRANSCRIPTION ON ACCENTED CORPUS

In this section, we test our model, GPT4-o (OpenAI, 2024) speech API, and SSDM (Lian et al., 2024) on three accented corpora: VCTK (Yamagishi et al., 2019), Common Voice (English) (Ardila et al., 2019), and GLOBE (Wang et al., 2024). VCTK includes speech data uttered by 109 native speakers of English with various accents. We randomly select approximately 20 speakers ( 10 hours) for inference. Common Voice contains 3,347 hours of audio from 88,904 speakers, recorded at a 48kHz sample rate. We only consider the English portion and randomly select 100 speakers ( 1 hour) with diverse accents (20 accents) for inference. GLOBE is recorded from 23,519 speakers at 24kHz, totaling 535 hours. We also randomly select 10 hours (20 accents) for inference. Note that Common Voice contains more noise than VCTK and GLOBE. Following Appendix A.15, we report False Positives from models. Unlike with fluent speech, accented speech can be considered dysfluent to some extent if the detected dysfluency type is *phonetic error*. Thus, we cannot say that the ground truth FP is zero, so we leave it blank. In addition to FP, we also report *phonetic pronunciation error rate* (PPER), which is computed by dividing the number of utterances where phonetic errors are detected (counted as one even when the number of errors exceeds 1) by the total number of samples. Evaluating the results using only FP and PPER presents challenges. Some predicted false positives or phonetic errors might exactly match the accents, meaning they are not

necessarily undesirable (i.e., lower values are not always better). Since we lack ground truth accent labels and human evaluation is prohibitively expensive, we employ a heuristic method: We measure the overlap between FP and PPER. The intuition is that the closer FP and PPER values are, the more likely the predicted phonetic errors match actual accents. Therefore, we define *Ratio* as *PPER* divided by *FP*. Results reported in Table 10 indicate that all models—GPT-4-o, SSDM, and SSDM 2.0—can predict both non-existent dysfluencies and dysfluencies corresponding to accents. Based on our heuristic evaluation methods, SSDM 2.0 appears to predict most accents accurately. However, determining the true false positive rate remains challenging and is left for future work.

Table 10: Evaluation on Accented Speech

| Eval Data | VCTK | | | Common Voice | | | GLOBE | | |
|---|---|---|---|---|---|---|---|---|---|
| | FP (%) | PPER (%) | Ratio (%, ↑) | FP (%) | PPER (%) | Ratio (%, ↑) | FP (%) | PPER (%) | Ratio (%, ↑) |
| Ground Truth | - | - | - | - | - | - | - | - | - |
| GPT4-o (OpenAI, 2024) | 11.0 | 4.3 | 39.1 | 17.2 | 5.4 | 31.4 | 17.4 | 5.0 | 28.7 |
| SSDM w/ Curri (Lian et al., 2024) | 17.7 | 10.3 | 58.2 | 23.9 | 15.0 | 62.8 | 18.2 | 14.2 | 78.1 |
| SSDM 2.0 (Ours) | 11.9 | 8.7 | **73.1** | 16.5 | 11.4 | **69.0** | 15.9 | 13.2 | **83.0** |

## A.17    REAL-WORLD STUTTERED SPEECH EVALUATION

In addition to nfvPPA speech, we also explored other real-world stuttered speech data. Following Yolo-Stutter (Zhou et al., 2024b), we performed zero-shot inference on two stuttering datasets: UCLASS (Howell et al., 2009) and **SEP-28K** (Bayerl et al., 2022a). **SEP-28K** is a large-scale dataset containing 28,177 clips extracted from public podcasts, though we excluded clips marked with "unsure" annotations. While UCLASS contains recordings from 128 stuttering speakers (both children and adults), we could only utilize 25 files due to annotation availability. Additionally, since these files lack block class annotations, we maintained consistency by excluding this class across all datasets. We evaluated both dysfluency type and timing, using manual annotations from Zhou et al. (2024b) for evaluation. Since both UCLASS and SEP-28K primarily contain repetition, prolongation, and block as dysfluency types, we included these in our evaluation. For timing assessment, we followed the *Time F1* metric proposed in Zhou et al. (2024b). Results shown in Table 11 demonstrate that SSDM 2.0 achieves state-of-the-art performance under all settings.

Table 11: Type-specific accuracy (ACC) and time F1-score

| Methods | Dataset | Accuracy (%, ↑) | | | Time F1 (↑) |
|---|---|---|---|---|---|
| | | *Rep* | *Prolong* | *Block* | |
| Kourkounakis et al. (Kourkounakis et al., 2021) | UCLASS | 84.46 | 94.89 | - | 0 |
| Jouaiti et al. (Jouaiti & Dautenhahn, 2022b) | UCLASS | 89.60 | 99.40 | - | 0 |
| H-UDM (Lian & Anumanchipalli, 2024) | UCLASS | 75.18 | - | 50.09 | 0.700 |
| YOLO-Stutter (Zhou et al., 2024b) | UCLASS | 92.00 | 91.43 | 56.00 | 0.893 |
| SSDM (Lian et al., 2024) | UCLASS | 92.00 | 91.70 | 60.08 | 0.898 |
| SSDM 2.0 (Ours) | UCLASS | **92.60** | **92.00** | **64.78** | **0.904** |
| Jouaiti et al. (Jouaiti & Dautenhahn, 2022b) | SEP-28K | 78.70 | 93.00 | - | 0 |
| H-UDM (Lian & Anumanchipalli, 2024) | SEP-28K | 70.99 | - | 66.44 | 0.699 |
| YOLO-Stutter (Zhou et al., 2024b) | SEP-28K | 82.01 | 89.19 | 68.09 | 0.813 |
| SSDM (Lian et al., 2024) | SEP-28K | 84.08 | 92.33 | 69.99 | 0.818 |
| SSDM 2.0 (Ours) | SEP-28K | **86.77** | **93.44** | **70.02** | **0.830** |

## A.18    ABLATIONS OF EACH MODULES AND LOSS FUNCTIONS

Here we detail the ablations of each module and loss function. SSDM 2.0 has introduced three major modules (NAF, FCSA, NICL) and multiple loss objectives for each module, making it necessary to explore the importance of each component. While Table 3 discussed the results when replacing individual SSDM (Lian et al., 2024) modules, here we present additional ablation studies. For simplicity, we focus on Libri-Dys inference experiments. We first explain the notation. Starting with the baseline SSDM (Lian et al., 2024), single-module replacements are denoted as: *SSDM+NAF*, where we replace SSDM's gestural scores with our NAF gestural scores; *SSDM+FCSA*, where we replace SSDM's CSA with our FCSA; and *SSDM+NICL*, where we replace SSDM's vanilla language modeling with our NICL. We can also replace multiple modules simultaneously: *SSDM+NAF+FCSA*, *SSDM+NAF+NICL*, and *SSDM+FCSA+NICL*. Note that SSDM 2.0 is equivalent to *SSDM+NAF+FCSA+NICL*. For loss objective ablations, we refer to the

complete loss function in Eq. 19. The NAF module involves three losses: $\lambda_1 \mathcal{L}_{\text{KL}} + \lambda_2 \mathcal{L}_{\text{FLOW}} + \lambda_6 \hat{\mathcal{L}}_{\text{PIT}}$, where only $\hat{\mathcal{L}}_{\text{PIT}}$ can be ablated as the first two are essential. FCSA includes two losses: $\lambda_3 \mathcal{L}_{\text{PRE}} + \lambda_4 \mathcal{L}_{\text{POST}}$, each of which can be ablated. NICL has a single loss $\lambda_5 \mathcal{L}_{\text{CON}}$ that can be ablated. These ablations are denoted as *SSDM+NAF-$\hat{\mathcal{L}}_{\text{PIT}}$*, *SSDM+FCSA-$\mathcal{L}_{\text{PRE}}$*, *SSDM+FCSA-$\mathcal{L}_{\text{POST}}$*, and *SSDM+NICL-$\mathcal{L}_{\text{CON}}$*.

All results are presented in Table 12. In terms of both F1 score and Matching Score (MS), replacing any single module in SSDM leads to performance improvement. Replacing an additional module (two modules in total) further enhances performance. Regarding the loss function, the posterior interpretable training (PIT) loss appears to have minimal influence. An interesting observation with FCSA is that incorporating both losses, $\mathcal{L}_{\text{PRE}}$ and $\mathcal{L}_{\text{POST}}$, delivers strong performance. However, removing either one results in a performance drop, although this trend becomes less pronounced with more data available. Overall, each module and each loss in our proposed framework demonstrates its effectiveness. For scalability, when all components are integrated, scalability increases dramatically. However, using only one or two modules yields less effective scalability improvements.

$$\mathcal{L}_{\text{FINAL}} = \lambda_1 \mathcal{L}_{\text{KL}} + \lambda_2 \mathcal{L}_{\text{FLOW}} + \lambda_3 \mathcal{L}_{\text{PRE}} + \lambda_4 \mathcal{L}_{\text{POST}} + \lambda_5 \mathcal{L}_{\text{CON}} + \lambda_6 \hat{\mathcal{L}}_{\text{PIT}} \tag{19}$$

Table 12: Detailed Ablations on Libri-Dys Dysfluency Detection

| Method | Eval Data | F1 (%, ↑) | MS (%, ↑) | F1 (%, ↑) | MS (%, ↑) | F1 (%, ↑) | MS (%, ↑) | F1 (%, ↑) | MS (%, ↑) | SF1 (%, ↑) | SF2 (%, ↑) |
|---|---|---|---|---|---|---|---|---|---|---|---|
| Training Data | | *LibriTTS (100%)* | | *Libri-Dys (30%)* | | *Libri-Dys (60%)* | | *Libri-Dys (100%)* | | | |
| SSDM (Lian et al., 2024) | Libri-Dys | 79.0 | 69.4 | 79.3 | 69.8 | 80.6 | 69.9 | 81.4 | 70.4 | 0.76 | 0.19 |
| w/o LLaMA | Libri-Dys | 78.3↓ | 69.0↓ | 78.8↓ | 69.2↓ | 79.6↓ | 69.3↓ | 80.7↓ | 70.0↓ | 0.65 | 0.25 |
| w/ Curri | Libri-Dys | 79.4 | 69.5 | 79.4 | 69.9 | 81.0 | 70.5 | 81.6 | 71.0 | 0.82 | 0.39 |
| SSDM+NAF (Ours) | Libri-Dys | 79.1 | 69.7 | 79.3 | 70.0 | 81.2 | 70.8 | 83.0 | 71.2 | 1.30 | 0.44 |
| -$\hat{\mathcal{L}}_{\text{PIT}}$ | Libri-Dys | 79.0 | 69.7 | 79.2 | 70.1 | 81.2 | 70.7 | 82.8 | 71.2 | 1.28 | 0.39 |
| +FCSA | Libri-Dys | 79.3 | 70.2 | 79.6 | 70.3 | 81.5 | 71.1 | 83.3 | 71.6 | 1.30 | 0.47 |
| +NICL | Libri-Dys | 79.6 | 70.0 | 79.7 | 70.5 | 81.6 | 71.3 | 83.5 | 71.8 | 1.33 | 0.47 |
| SSDM+FCSA (Ours) | Libri-Dys | 79.3 | 69.7 | 79.7 | 70.2 | 80.9 | 70.2 | 81.7 | 70.9 | 0.72 | 0.21 |
| -$\mathcal{L}_{\text{PRE}}$ | Libri-Dys | 79.0 | 69.3 | 79.4 | 70.0 | 80.4 | 69.9 | 81.3 | 70.4 | 0.67 | 0.11 |
| -$\mathcal{L}_{\text{POST}}$ | Libri-Dys | 79.0 | 69.5 | 79.5 | 70.1 | 80.6 | 70.2 | 81.6 | 70.8 | 0.74 | 0.22 |
| +NICL | Libri-Dys | 79.4 | 70.0 | 79.8 | 70.6 | 81.2 | 70.7 | 82.0 | 71.3 | 1.20 | 1.00 |
| SSDM+NICL (Ours) | Libri-Dys | 79.2↑ | 69.9↑ | 79.3 | 69.9↑ | 81.9↑ | 71.9↑ | 82.8↑ | 72.4↑ | 1.31 | 0.95 |
| -$\mathcal{L}_{\text{CON}}$ | Libri-Dys | 79.0 | 69.5 | 79.0 | 69.4 | 81.5 | 71.6 | 82.3 | 72.1 | 1.24 | 1.03 |
| SSDM 2.0 (Ours) | Libri-Dys | **79.9** | **70.3** | **80.0** | **70.3** | **83.2** | **73.4** | **86.2** | **75.9** | **2.18** | **1.99** |

# B  RELATED WORK

**Dysfluency Modeling**  Speech dysfluency modeling seeks to detect dysfluencies at both word and phoneme levels, with precise timing given a reference text (Lian et al., 2023b). Early work focused on hand-crafted features (Chia Ai et al., 2012; Chee et al., 2009; Esmaili et al., 2016; Jouaiti & Dautenhahn, 2022a; Mujtaba et al., 2024) and end-to-end classification approaches at both utterance level (Kourkounakis et al., 2021; Alharbi et al., 2020; Jouaiti & Dautenhahn, 2022b; Oue et al., 2015; Bayerl et al., 2022b; Howell & Sackin, 1995; Alharbi et al., 2017; Tan et al., 2007; Bayerl et al., 2023a;b; Dash et al., 2018; Mohapatra et al., 2023; Wagner et al., 2024; Changawala & Rudzicz, 2024) and frame level (Shonibare et al., 2022; Harvill et al., 2022). The current mainstream methods treat this problem as a time-based object detection task (Lian et al., 2023b; Lian & Anumanchipalli, 2024; Zhou et al., 2024b;a; Lian et al., 2024). More recently, token-based methods (Zhou et al., 2024c) have also been explored and have achieved comparable results.

**Articulatory Speech Representation Learning**  Recent studies show articulatory features' effectiveness as scalable representations in speech recognition (Lian et al., 2023a) and dysfluency modeling (Lian et al., 2024). Early research sought to resolve speech dynamics through motion laws (Coker, 1976), simplified by gestural theory (Browman & Goldstein, 1990; 1992) which conceptualizes speech as sparse activations of articulatory primitives, analogous to robotics' gait libraries (Grizzle et al., 2010). Subsequent work developed methods for automatic gesture extraction (Ramanarayanan et al., 2013; Lian et al., 2022; 2023a). Recently, articulatory-to-speech inversion (Cho et al., 2024) enable extraction of articulatory-free representations as speech codecs with full intelligibility, validated as optimal encodings for dysfluency modeling (Lian et al., 2024).

