# OpenReview forum: "Time-Accurate Speech Rich Transcription with Non-Fluencies"
_ICLR.cc/2025/Conference — ICLR 2025 Conference Withdrawn Submission_

### Official Review · Reviewer_cH6N · 2024-10-27

**Soundness:** 3
**Presentation:** 3
**Contribution:** 2
**Rating:** 5
**Confidence:** 3

**Summary:**

The paper presents a novel pipeline to model and identify speech disfluencies. In tandem it presents a sizable dataset containing disfluent utterances for further work in the field.

**Strengths:**

The paper offers several novel techniques to assist in ongoing work for modeling speech disfluencies. I particularly find their development of the neural articulatory flow model fruitful for in-depth modeling of patient vocalizations during therapy. Each method utilized in their pipeline is of significant interest for a standalone paper on the topic, and has sizable justification for consideration in work in this field. In addition, their development of a dysfluency corpus is incredibly important for the field of dysfluency modeling (most datasets are rather sparse and limited in coverage, there is a critical need in the research community for more datasets for study).

**Weaknesses:**

This paper has a significant organizational issue that makes it difficult for an outside reader to follow. The vast majority of necessary information for a researcher outside disfluent speech research to understand the paper has been relocated to the appendix. This makes reading and evaluation of the paper rather difficult. For example, the researchers have chosen nfvPPA as a primary case study for their experiments in a field that could just as well consider Parkinsons and other variants of Aphasia as a case study. This information is not presented to the reader in the paper proper; it is instead moved to Appendix A.2. This is information that should be more clearly presented.

While this would not be an issue for a specialized conference, ICLR is general enough that consideration should be made for outside researchers to follow. Care should be made for establishing clear cases of dysfluency and conditions that cause it. As well, given that articulation modeling is a key subject of the paper, it should be presented clearly in the background section.

**Questions:**

Suggestion: reconsider what elements in the main paper and which elements in the appendix should be moved. Please take consideration that your colleagues may not have knowledge of dysfluency research or even models of language production.

Could you clarify why your models consider codecs? If was difficult to follow why these should serve as the audio input format.

Have you considered the potential bias involved in evaluating speech production and how it would dovetail with use of Speech LMs? While in aphasic cases this is more clear cut, I'm suspicious about how the work may mislabel dialect speakers as dysfluent speakers.

---

> ### Author Response · Authors · 2024-11-19
>
> We thank the reviewers for their in-depth reviews. We provide our rebuttal below, and we have updated our manuscript with the changes highlighted in yellow.
>
> 1. Organization issue. (Definition of dysfluency, nfvPPA, etc)
>
> We appreciate the suggestion to make a more general case for dysfluent speech, and specifically nfvPPA in the main paper. To make it clearer and more understandable for all audiences, we have reorganized the introduction by adding discussions about motivations, dysfluency, and nfvPPA at the beginning. Here is our logic (which is also reflected in the updated paper with yellow highlights).
>
> **We first introduce dysfluency and its relevance, even in general purpose ASR or conversational analysis, and conclude that disordered speech represents an extreme but generalizable case that we primarily focus on in this work**.
>
> > Current automatic speech recognition (ASR) [1] or speech language models (SLMs) [2] mostly transcribe speech to text while ignoring *dysfluency* (dysfluency, non-fluency, disfluency are interchangeable), which includes *repetition, deletion, insertion, replacement of sounds, filler words, or hesitations* [3]. Accurately transcribing dysfluencies has three major applications:
> >
> >> (i). In spontaneous or daily conversational speech, dysfluencies help understand speaker intent and conversational behavior;
> >
> >> (ii). For accented speakers and language learners, dysfluencies provide effective feedback for pronunciation improvement;
> >
> >> (iii). For people with speech disorders like Aphasia, dysfluency transcription could reduce speech language pathologists' workload and aid diagnosis.
> >
> > Since the first two cases involve near-fluent speech, dysfluent speech transcription serves as a fundamental method in both typical and atypical speech recognition, extending beyond clinical applications and is a **superset** of the ASR problem. Looking at the interface, ASR models typically output only **what the person intended to say**, while its superset, dysfluency transcription, provides two outputs: **what the person intended to say** and **what the person actually said**. For example, *Please call stella* and *P-Please c-call st-ah-lla*. Actually, a significant portion of dysfluencies arise in disordered speech only. By accurately transcribing these, we can naturally extend this capability to less severe cases, such as conversational or accented speech. Thus, we prioritize disordered speech dysfluencies as a foundation, which can then be extended to a superset of ASR problem.
>
> We then highlight the relevance of **a common set of behavioral markers (dysfluencies) across various speech disorders**, emphasizing the potential for developing a unified framework that addresses all speech disorders. Following this, we discuss our rationale for focusing on nonfluent primary progressive aphasia (nfvPPA) as our initial target. Finally, we provide an overview of nfvPPA, including its defining characteristics and clinical significance.
>
> > Speech disorders are accompanied by various symptoms and can be associated with conditions such as dyslexia, aphasia, apraxia, Parkinson's disease, stuttering, dysarthria, cerebral palsy, autism spectrum disorder (ASD), stroke, and other neurological conditions. Testing all these disorders is not practical due to data  constraints. **Different disorders also present varying levels of neurological conditions and dysfluency behaviors (severity). However, they all share the same set of dysfluencies as mentioned before**. Thus, **we aim to develop a general dysfluency transcription system that can be used by all disorders for follow-up diagnosis correspondingly**.
>
> > In this work, we perform inference on real-world data including open-source stuttering data and  nonfluent variant primary progressive aphasia (nfvPPA, [4]). nfvPPA uniquely presents with pronounced speech dysfluency, characterized by deficits in syntax and motor speech (apraxia of speech), making it an ideal candidate for testing the pipeline.
> In summary, __while disorders may have different neurological or physiological causes, dysfluency transcription captures their shared behavioral symptoms (dysfluencies)__, and thus we only focus on how to accurate transcribe the dysfluencies in this work. “
>
> __Please feel free to review our updated manuscript highlighted in yellow__ and we would also greatly appreciate any suggestions from you on how to make these concepts (dysfluency, etc) more accessible to the broader research community.
>
> [1] Radford, Alec, et al. "Robust speech recognition via large-scale weak supervision." ICML 2023.
>
> [2] Haibin Wu, et al. Towards audio language modeling-an overview. arXiv preprint arXiv:2402.13236, 2024.
>
> [3] Lian, Jiachen, et al. "Unconstrained dysfluency modeling for dysfluent speech transcription and detection." ASRU, 2023.
>
> [4] Gorno-Tempini, Maria Luisa, et al. "Classification of primary progressive aphasia and its variants." Neurology 76.11 (2011): 1006-1014.

---

> > ### Comment · Reviewer_cH6N · 2024-11-19
> >
> > Presentation at beginning is much better. May improve clarity to also use terms like 'literal' and 'standard' speech. I've seen these terms used for equivalent discussions so may assist communication with wider audience.
> >
> > Further clarity can be made my focusing on articulation based disfluency (Parkinsons, Broca's) as opposed to semantic based processing issues (Wernicke's, ASD). Your case is strong enough that the inclusion of L2 speakers/learners may be unnecessary. Narrowing domain to pathological speech disorders avoids potential confusion.
> >
> > Related paper (reviewer does not require it in literature review): https://arxiv.org/abs/2402.08021

---

> ### Author Response · Authors · 2024-11-19
>
> 2. Care should be made for establishing clear cases of dysfluency and conditions that cause it
>
> This is a truly in-depth comment. Currently, we focus only on transcribing dysfluencies, which are common behavioral symptoms across all speech disorders. We leave diagnosis either for human effort as a next step or for our future work. However, we can discuss this further.
> >
> >> (1) In collaboration with clinical partners, we recognize that understanding the underlying causes of specific disorders remains an open problem. This investigation might require exploring brain data or conducting joint brain-speech analyses. Different disorders, or even different subjects within the same disorder, can have different causes. **These causes can be both neurological (brain) or physiological (articulatory) conditions**.
> >
> >> (2) Computational methods such as our dysfluency transcriber could help **establish relevant biomarkers at the behavioral level**, which is an easier step, rather than at the neurological or physiological levels, which are harder to evaluate. One approach would be to quantify different types of dysfluencies and analyze their patterns as potential biomarkers. For instance, nfvPPA might be associated with increased missing or insertion frequencies across specific age groups. We believe our framework has the potential to enhance such diagnostic capabilities.
> >
> > Following your suggestion, we plan to include case studies in our future work. We would greatly appreciate your additional thoughts and suggestions on this direction.

---

> ### Author Response · Authors · 2024-11-19
>
> 3. Articulation modeling.
>
> This is also the concern from reviewer i1kS, so we unify the reply, and __already added more introduction in Sec. 3.1 and Section 3 (before 3.1), highlighted via green in our updated manuscript__. Feel free to check. And we detail them here too:
>
> > Current speech representation modeling typically uses explicit $D \times T$ matrices, where T represents time and D represents channel dimension, and these are learned densely in a data-driven manner. However, human speech is produced by a sparse set of articulators with sparse activation in time. If we define basic moving patterns of articulators as a dictionary and project articulatory data into this dictionary space, we obtain a sparse activation matrix. The dictionary is called _gestures_ and the sparse activation matrix is called _gestural scores_ [1]. This motivates us to ask: instead of densely learning elementwise speech representations, can we sparsely and implicitly learn such structural speech representations via human-prior rules?
>
>
>
> > In this work, we propose _Neural Articulatory Flow_, which involves a _Semi-implicit Speech Encoder_ (Section 3.1) to predict only the indices of active regions in articulation, and an _Articulatory Flow_ (Section 3.2) to distill speech intelligibility from pretrained acoustic speech embeddings (WavLM [2]). We also optionally introduce _Interpretable Posterior Training_ to visualize how speech is physically produced in articulation and to derive articulatory feedback for pronunciation.
>
> > Given a speech waveform, we adopt _UAAI_ (**U**niversal **A**coustic-to-**A**rticulatory **I**nversion) [3] to obtain its articulatory trajectory $X\in \mathbb{R}^{D\times T}$ at 50Hz. UAAI consists of a WavLM [2] encoder followed by a linear layer, which takes speech waveform as input and predicts articulatory trajectories. The model was trained on the MNGU0 corpus [4], which contains 75 minutes of EMA (Electromagnetic midsagittal articulography) data collected during newspaper reading. In our setting, $D=12$ corresponds to the x and y coordinates of 6 articulators measured in the EMA recording process.
> We define articulatory trajectory kernels (_gestures_) $G^{T'\times D\times K}$, where $T'$ is the window size and $K$ is the number of kernels. By projecting $X$ onto the kernel space, we obtain _gestural scores_ $H\in \mathbb{R}^{K\times T}$, which is high-level abstraction of $X$. $H$ denotes when the articulators are activated for how long. In this work, we directly predict $H$ from $X$ without _gestures_ $G$ for simplicity, focusing only on active indices that indicate articulatory activation with a _Count Encoder_, a _Index Encoder_ and a _Value Encoder_. We provide a visualization of gestural scores $H$ and its correlation with $X$ and _Gestures_ $G$ in Appendix. A.5. “
>
> Feel free to let us know if you want more content to appear in the main body.
>
> [1] Catherine P Browman and Louis Goldstein. Articulatory phonology: An overview. Phonetica, 49(3-4):155–180, 1992.
>
> [2] ​​Sanyuan Chen, Chengyi Wang, Zhengyang Chen, Yu Wu, Shujie Liu, Zhuo Chen, Jinyu Li, Naoyuki Kanda, Takuya Yoshioka, Xiong Xiao, et al. Wavlm: Large-scale self-supervised pre-training for full stack speech processing. IEEE Journal of Selected Topics in Signal Processing, 16(6):1505–1518, 2022.
>
> [3] Cheol Jun Cho, Peter Wu, Tejas S Prabhune, Dhruv Agarwal, and Gopala K Anumanchipalli. Articulatory encodec: Vocal tract kinematics as a codec for speech. arXiv preprint arXiv:2406.12998,2024
>
> [4] Korin Richmond, Phil Hoole, and Simon King. Announcing the electromagnetic articulography (day1) subset of the mngu0 articulatory corpus. In Twelfth Annual Conference of the International Speech Communication Association, 2011.

---

> ### Author Response · Authors · 2024-11-19
>
> 4. Could you clarify why your models consider codecs?
>
> > Our work involves two different 'codecs.'
> >
> >> (1) The first is related to Universal Articulatory-to-Acoustic Inversion (UAAI) codec model [1], which generates articulatory trajectories (X) from speech. Since our articulatory modeling requires speech's articulatory trajectory, which is typically unavailable in real speech data, we use offline UAAI to synthesize X. The UAAI work we referenced demonstrates that speech can be converted to articulatory trajectories which, combined with other speaker features, can resynthesize highly intelligible speech. This suggests that articulatory trajectories serve a similar role to traditional neural speech codecs like SoundStorm, and thus, it was called articulatory codec models.
> >
> >> (2) The second codec is SoundStorm [2], which we used in our experiments (Table 1). Our motivation was to test whether our proposed gestural scores are more scalable than neural speech codecs, and our current findings suggest they are.
> >
> [1] Cheol Jun Cho, Peter Wu, Tejas S Prabhune, Dhruv Agarwal, and Gopala K Anumanchipalli. Articulatory encodec: Vocal tract kinematics as a codec for speech. arXiv preprint arXiv:2406.12998,2024
>
> [2] Zal´an Borsos et al. Soundstorm: Efficient parallel audio generation. arXiv preprint arXiv:2305.09636, 2023.

---

> > ### Comment · Reviewer_cH6N · 2024-11-20
> >
> > I don't believe the comparison to be well motivated and bridging the issue makes for a weaker work. If you want to consider codec comparisons, then conclusive results would need to also consider competitors such as Encodec (https://arxiv.org/abs/2210.13438) along with the typical comparison against HuBERT/w2v-BERT/wav2vec style embeddings. You would also need to address the null case of real speech data for simple benchmarking. I'd suggest removing the comparison for a more cohesive argument (there is enough novelty in the paper that streamlining can be afforded).

---

> ### Author Response · Authors · 2024-11-19
>
> 5. Regarding bias.
>
> We believe one primary bias stems from the distribution gap between training and inference data, as the model typically tends to overfit to the training data distribution. A significant aspect of this bias is **accent variation**. Our best-performing model was trained on Liri-Co-Dys, which was simulated based on LibriTTS. This dataset primarily contains United States American accents and does not cover a wide range of other accents. This limitation raises two questions.
>
> > (1) **Can our model, trained on US accents, generalize to accented data** from datasets like VCTK, Common Voice [1], and GLOBE [2] for dysfluency detection?
> >
> >> To answer this, we evaluate GPT4-o speech API, SSDM, and our proposed SSDM 2.0 in zero-shot dysfluency detection across three accented corpora:
> >
> >> VCTK contains speech data from 109 native English speakers with various accents. We randomly selected approximately 20 speakers (~10 hours) for inference.
> >
> >> Common Voice comprises 3,347 hours of audio from 88,904 speakers, recorded at 48kHz. For our evaluation, we focused on the English portion, randomly selecting 100 speakers (~1 hour) representing 20 diverse accents.
> >
> >> GLOBE contains recordings from 23,519 speakers at 24kHz, totaling 535 hours. We randomly selected 10 hours of data covering 20 accents for inference.
> >
> >> Note that Common Voice exhibits higher noise levels compared to VCTK and GLOBE.
>
>
> > (2) **Is accent dysfluency or not**?
> >
> >> This question is crucial for evaluating accented speech. The definition of accent is broad and not unified, encompassing pronunciation variations, stress and intonation variations, and speaking rate (rhythm) variations. In SSDM 2.0, **we consider phoneme replacement or phonetic error, which significantly overlaps with "pronunciation variations."** Therefore, if SSDM 2.0 predicts a phonetic error that aligns with accent-related variations, this should be considered an accurate prediction rather than a true false positive. This means that using false positives (FP) as an evaluation metric can sometimes yield desired values. Since accented speech can be considered partially dysfluent when the detected dysfluency type is _phonetic error_, we cannot assume the ground truth FP is zero, and thus leave it blank.
> >
> >> In addition to FP, we introduce _phonetic pronunciation error rate_ (PPER), calculated by dividing the number of utterances containing detected phonetic errors (counted as one regardless of multiple errors) by the total number of samples.
> Evaluating results using only FP and PPER presents challenges. Some predicted false positives or phonetic errors may precisely correspond to accents, making them not necessarily undesirable (i.e., lower values aren't always better). Given the lack of ground truth accent labels and the prohibitive cost of human evaluation, **we employ a heuristic method: measuring the overlap between FP and PPER. The intuition is that closer FP and PPER values suggest predicted phonetic errors more likely match actual accents**. We define _Ratio_ as _PPER_ divided by _FP_.
> >
> >> Results in Table-12 show that all models—GPT-4-o, SSDM, and SSDM 2.0—can predict both non-existent dysfluencies and dysfluencies corresponding to accents. Based on our heuristic evaluation methods, **SSDM 2.0 appears to predict most accents accurately**. However, determining the true false positive rate remains challenging and is left for future work.
>
> **Please refer to Appendix A.16 (highlighted in yellow) and Table 10 in our updated manuscript for detailed results.**
>
> [1] Ardila, Rosana, et al. "Common voice: A massively-multilingual speech corpus." arXiv preprint arXiv:1912.06670 (2019).
>
> [2] Wang, Wenbin, et al. "GLOBE: A High-quality English Corpus with Global Accents for Zero-shot Speaker Adaptive Text-to-Speech." arXiv preprint arXiv:2406.14875 (2024).
>
> * Another interesting direction, as suggested by Reviewer i1kS, is testing our models on fluent speech. This evaluation should yield lower false positives (FP) and better word error rate (WER). For ASR tasks, we use Whisper V2 as a baseline and GPT4-o real-time speech API for testing false positives. We first evaluate performance in a zero-shot setting on the LibriTTS corpus. We then fine-tune our model on LibriTTS with target dysfluency labels set to "None" (denoted as SSDM 2.0-Tuned), with **results shown in Table 11 (updated)**.
> Results indicate that Whisper achieves the best ASR performance due to its scaling efforts. While GPT4-o speech real-time interface produces some false positives on fluent speech, SSDM shows higher FP and worse WER scores. SSDM 2.0 demonstrates better FP rates than GPT4-o but falls short of Whisper's ASR performance. After "fluent" fine-tuning, SSDM 2.0-Tuned achieves the best FP scores and ASR performance comparable to Whisper. **For detailed results, please refer to Appendix A.11 and Table 9 (highlighted in red) in our updated manuscript.** We welcome any additional feedback!

---

> ### Author Response · Authors · 2024-11-21
>
> Thank you for your valuable new suggestions. Here is our reply
>
> > 1. We appreciate the suggestion and believe 'lexicalized speech' (the words that were spoken) and 'uttered speech' (how the words were spoken) to be more accurate statements. Thus, we adopt them in the paper. Thanks for pointing them out!
> >
> > 2. We agree that disordered speech modeling is a stronger case than L2 language learning, so we have removed this along with conversational speech dysfluencies. We focus solely on pathological speech. Since nfvPPA is indeed associated with articulatory-based dysfluency, we make our statement as follows:
> >
> >> * Current automatic speech recognition (ASR) systems and speech language models (SLMs) typically transcribe speech into _the words that were spoken (lexicalized speech)_ rather than _how the words were spoken (uttered speech)_. For example, when someone says _P-Please c-call st-ah-lla_, these systems usually perform _denoising_ and output _please call stella_. This approach is suitable for most spoken dialogue scenarios and services, and helps reduce confusion in communications. However, in _pathological speech_ domain, _uttered speech_ is required to accurately identify articulation and pronunciation problems for diagnostic purposes. The gap between lexicalized speech and uttered speech is referred to as _dysfluency_, which includes _repetition, deletion, insertion, replacement of sounds, filler words, and hesitations_. Accurate transcription of speech dysfluencies could substantially reduce the workload of speech language pathologists while facilitating diagnosis and serving as a powerful clinical assessment tool.
> >
> >> * Pathological speech disorders are typically caused by neurological or physiological factors and are associated with various dysfluencies, such as motor and phonological (articulatory) dysfluencies (e.g., nfvPPA, Parkinson’s disease, Broca’s aphasia) and higher-order (or semantic) dysfluencies (e.g., svPPA, Wernicke’s aphasia, ASD). Diagnosing these disorders is challenging due to case-by-case variability and differences in severity. However, they share a common set of dysfluencies at the behavioral level, making it possible to develop a general dysfluency transcription system that can accommodate all disorders and support follow-up diagnosis. Due to data constraints, however, it is not feasible to test such a system on all pathological speech data. Therefore, we focus on nfvPPA, leveraging the currently available data to develop a dysfluency transcription tool specifically targeting _articulation-based dysfluencies_.
> >
> >> We also emphasized _articulatory dysfluencies_ in abstract.
> >
> >> In Section 7, we discussed that as future work, other articulatory-based disorders and semantic-based disorders will be explored.
> >
> > 3. In Section 3, we also mentioned that learning articulatory representations is a natural way to model articulatory dysfluencies.
> >
> > 4. Your suggestion regarding codec makes sense. The other insight is that codec is more acoustic-focused while dysfluency representations would be more semantic-focused. Using SSL units is sufficient, so we have removed the codec.
> >
> >
> We really appreciate your high-quality review. **Please check our updated manuscript again (highlighted in deeper yellow than before)**. Feel free to let us know if you have any other questions or suggestions

---

> > ### Comment · Reviewer_cH6N · 2024-11-23
> >
> > Final changes satisfy my comments

---

> > > ### Author Response · Authors · 2024-11-23
> > >
> > > Thank you for your thoughtful comments and for acknowledging that the final changes have addressed your concerns. We are happy to provide any additional clarifications or adjustments if needed. If you feel that the paper now meets the criteria for a stronger recommendation, we would greatly appreciate your consideration of an updated score.

---

> ### Author Response · Authors · 2024-11-27
>
> Dear reviewer
>
> Thank you for your constructive feedback, particularly regarding the paper's presentation. And sorry for bothering you at this time. We apologize for any additional effort on your part. If **we have addressed the identified concerns**, we understand that according to recent new PC regulations, we are eligible to respectfully request a score raising. Beyond the writing problems only, if you have any additional concerns regarding the **technical contributions (algorithms, models, etc.)**, we would be happy to clarify and discuss those as well.
>
> Sincerely

---

### Official Review · Reviewer_f1SE · 2024-11-03

**Soundness:** 3
**Presentation:** 3
**Contribution:** 2
**Rating:** 5
**Confidence:** 2

**Summary:**

This paper proposes a framework named SSDM 2.0 to better transcribe the speech with dysfluencies. This framework includes Neural Articulatory Flow (NAF) to extract scalable and efficient dysfluency-aware speech representation, Full-Stack Connectionist Subsequence Aligner(FCSA) for dysfluency alignments, and Mispronunciation In-Context Learning and Consistency Learning in langauge
model for dysfluency transfer. On top of that, the author open-sources a large simulated speech co-dysfluency corpus called Libri-Dys-Co.

Experiments are designed to evaluate the scalability and efficacy of the proposed NAF representation and the performance of SSDM 2.0 in detecting dysfluencies in speech.

**Strengths:**

* Novel methodology. The paper proposes multiple new methodologies such as Neural Articulatory Flow which encodes semi-implicit speech representations that align closely with human speech production and Fullstack Connectionist Subsequence Aligner(FCSA) which develops a comprehensive alignment mechanism capable of handling diverse dysfluencies, improving the precision of non-fluency detection. The proposed methods are shown to significantly reduce computational complexity, allowing the model to scale effectively with larger datasets.
* Comprehensive evaluation. The author designs evaluation and benchmarks to cover proposed methods.
* Contribution of new resources. This paper open-sources a large-scale co-dysfluency corpus (Libri-Co-Dys) with over 6,000 hours of annotated speech, providing a valuable resource for future research in dysfluency transcription and detection.
* Potential real world application: The proposed framework could accurately transcribe and detect dysfluencies and has significant implications for speech therapy, language screening, and spoken language learning.

**Weaknesses:**

* The writing style of this paper comes with overly formal language and verbose phrasing. One instance is the beginning of section 3: "Notwithstanding, the neural gestural system proposed by Lian et al. (2024) is encumbered by complex design and challenges in equilibrating multiple training objectives such as self-distillation." The usage of "notwithstanding", "encumbered" and "equilibrate" is not necessary and they all have synonyms that are commonly used. The phrasing of the sentence is not concise. Making the paper overall less comprehensible to the readers.
*  Framework complexity. Even though the paper claims that SSDM 2.0 has 10x less training complexity of its precedent work SSDM, the integration of NAF, FCSA, and NICL adds more layers of complexity to the model. There is lack of complexity comparison of other types of baseline systems to provide a reference of where it stands in all viable options.
* The presentation of the paper needs to improve:
  + Many concepts mentioned in this paper are not self-contained and the reader has to refer to other papers. For instance, AAI.
  + Given the layers of complexity brought by multiple different modules/methods in the framework, an overview of the it should be first given based on Figure 2 before breaking it down to individual components.
* Lack of ablation studies. Table 1 only shows difference with or without neural articulatory flow loss and post-interpretable training, there is no assessment of the significance of other proposed components.
* Lack of real-world data. There is a over-reliance on synthetic data which arises questions about how the system performs in real world scenarios.

**Questions:**

+ I suggest removing the claim of complexity improvement from the paper if there is no enough space to explore in details.
+ In section 6.7, I think a more fair baseline would be SSDM 2.0 removing the NICL tuning steps.

---

> ### Author Response · Authors · 2024-11-19
>
> We thank the reviewers for their critical reviews regarding the writing style, ablation studies, and other concerns. While we have made extensive revisions to improve clarity, we welcome further feedback!
>
> 1. The reviewers noted that the paper's writing style was characterized by overly formal language and verbose phrasing.
> >
> > For the specific instances mentioned, we have rewritten the text to introduce gestures and our neural articulatory flow in a more accessible and easy-to-read manner. We have also added clarification for "self-contained" concepts such as "UAAI" and provided detailed information about its training data. **These revisions can be found in Section 3 (the text immediately following the title but preceding Section 3.1) and Section 3.1, which are highlighted in green in our updated manuscript. To enhance reader comprehension of Figure 2, we have added a new Section 2 that introduces the SSDM 2.0 architecture (highlighted in red)**.
> >
> > Additionally, as suggested by Reviewer i1kS, **we have expanded our explanation of gestures and gestural scores in Section 3.1, marked in red for easy reference**.
> >
> > Furthermore, following Reviewer cH6N's recommendation, we have reorganized the introduction to include more comprehensive background information about dysfluency and disordered speech processing, helping readers better understand our motivation. **These changes are highlighted in yellow in the Introduction section.**
> >
> > **Please feel free to review our updated manuscript** and let us know if you have additional suggestions for improving the writing style and clarity.
>
> 2. Regarding SSDM 2.0 Complexity.
>
> SSDM 2.0 incorporates different components (NAF, FCSA, NICL) which may appear complex, or as one reviewer noted, "adds more layers of complexity to the model." We address these concerns as follows:
> >
> > We evaluated five baselines for dysfluency transcription: SALMONN [1], LTU-AS [2], GPT-4 (speech), GPT-4o (speech), and SSDM [3]. As shown in Table 4, the first four speech language models (SLMs) perform significantly worse than SSDM and SSDM 2.0. Although models like SALMONN appear more concise, their weak performance makes them unsuitable for our task. Even after our fine-tuning SALMONN with our Libri-Co-Dys data (denoted as SALMONN-13B-FT in Table 4), improvements were minimal.
> >
> > When a simpler model yields poor performance while a more complex model demonstrates substantial improvements, the latter would become the more practical choice. Regarding model parameters, we follow LTU-AS [2] by **using only LLaMa-7B, which is significantly more efficient than SALMONN-13B**. Therefore, **SSDM serves as a more appropriate comparison point**.
> >
> > In SSDM, gestural scores are derived through complex processes including convolutional matrix factorization, sparse sampling, modeling, self-distillation, and phoneme classification. In contrast, our NAF employs only an encoder process and uses lightweight flow matching for self-distillation, making it substantially simpler than SSDM's gestural scores. Additionally, FCSA introduces no additional computational cost compared to SSDM's CSA.
> >
> > In summary, while SSDM 2.0 would be more efficient than traditional SLMs like SALMONN, it is more appropriate to compare it with SSDM. Importantly, SSDM 2.0 achieves greater efficiency than SSDM, particularly in gestural score modeling.
> >
> > Regarding complexity, **we primarily focus on training convergence rate and representation sparsity.** As summarized in Appendix A.14, our model achieves a 10X faster convergence rate compared to SSDM. Furthermore, as shown in Figure 3, our gestural scores (green dots) exhibit sparse representations (with sparsity around 0.9). This sparsity allows us to utilize sparse operations during backpropagation, which we believe contributes to the faster convergence rate. As per reviewer's request, **we have moved these discussions from the main paper** to the appendix for future exploration while removing them from the main paper. We welcome any additional feedback on these aspects!
>
> [1] Tang, Changli, et al. "Salmonn: Towards generic hearing abilities for large language models." ICLR, 2024
>
> [2] Gong, Yuan, et al. "Joint audio and speech understanding." 2023 IEEE Automatic Speech Recognition and Understanding Workshop (ASRU). IEEE, 2023.
>
> [3] Lian, Jiachen, et al. "SSDM: Scalable Speech Dysfluency Modeling." arXiv preprint NeurIPs, 2024

---

> ### Author Response · Authors · 2024-11-19
>
> 3. Lack of Real-World Data.
>
> We acknowledge our limited use of real-world data. There are two key constraints that distinguish dysfluent speech from fluent speech:
>
> > (1) Dysfluent speech primarily occurs in stuttered/disordered speech. **Acquiring such data involves ethical and privacy concerns that limit scalability**. For instance, the nfvPPA data we currently use required several months of IRB approval, and even then remains limited in scale.
>
> > (2) Both training and inference require **detailed dysfluency annotations**. Currently available real-world datasets like UCLASS [1] and SEP28K [2] only provide utterance-level class labels, which are insufficient for our needs. Additionally, both UCLASS and SEP28K are relatively small-scale datasets.
>
> Thus we did the following explorations:
>
> > (1) We relabeled UCLASS and SEP28K by adding time labels, where train/test split is consistent with YOLO-Stutter [3], and performed zero-shot inference on these stuttering datasets. SEP-28K is a large-scale dataset containing 28,177 clips from public podcasts. UCLASS contains recordings from 128 stuttering speakers.
> >
> >> We evaluated both dysfluency type and timing. As UCLASS and SEP-28K primarily contain repetition, prolongation, and block as dysfluency types, we focused our evaluation on these categories. For timing assessment, we used the _Time F1_ metric proposed in YOLO-Stutter. Results in Table 11 demonstrate that SSDM 2.0 achieves state-of-the-art performance across all settings. For detailed results, **please refer to Appendix A.17 (highlighted in green) and Table 11 in our updated manuscript**.
> >
> > (2) Accent speech (same reply with Reviewer cH6N). **Accent speech partially overlaps with dysfluent speech, particularly regarding phonetic pronunciation replacement**. We evaluated GPT4-o speech API, SSDM, and our proposed SSDM 2.0 in zero-shot dysfluency detection across three accented corpora:
> >
> >> VCTK [4] contains speech data from 109 native English speakers with various accents. We randomly selected approximately 20 speakers (~10 hours) for inference.
> >
> >> Common Voice [5] comprises 3,347 hours of audio from 88,904 speakers, recorded at 48kHz. For our evaluation, we focused on the English portion, randomly selecting 100 speakers (~1 hour) representing 20 diverse accents.
> >
> >> GLOBE [6] contains recordings from 23,519 speakers at 24kHz, totaling 535 hours. We randomly selected 10 hours of data covering 20 accents for inference.
> >
> > Note that Common Voice exhibits higher noise levels compared to VCTK and GLOBE.
> >
> > In SSDM 2.0, we consider phoneme replacement or phonetic error, which significantly overlaps with "pronunciation variations." When SSDM 2.0 predicts a phonetic error that aligns with accent-related variations, this should be considered an accurate prediction rather than a true false positive. Consequently, using false positives (FP) as an evaluation metric can sometimes yield desired values. We introduce _phonetic pronunciation error rate_ (PPER) in addition to FP. PPER is calculated by dividing the number of utterances containing detected phonetic errors (counted as one regardless of multiple errors) by the total number of samples. Given the lack of ground truth accent labels and the prohibitive cost of human evaluation, we employ a heuristic method: measuring the overlap between FP and PPER. The intuition is that closer FP and PPER values suggest predicted phonetic errors more likely match actual accents. We define _Ratio_ as _PPER_ divided by _FP_.
> >
> > Results in Table-12 show that SSDM 2.0 appears to predict most accents accurately. However, determining the true false positive rate remains challenging and is left for future work.
> >
> > **Please refer to Appendix A.16 (highlighted in yellow) and Table 10 in our updated manuscript for detailed results.**
> >
> > (3) Dysfluent speech in real-world scenarios primarily occurs in stuttered or disordered speech. Regarding our clinical speech data, we are continuing data collection and will report on more subjects in the future. Additionally, we plan to collect "human simulated" stutter data, including from our own simulations, as part of our future work.
>
> [1] P. Howell, et al, “The uclass archive of stuttered speech,” Journal of Speech Language and Hearing Research, vol. 52, p. 556, 2009.
>
> [2] C. Lea, V. Mitra, et al, “Sep-28k: A dataset for stuttering event detection from podcasts with people who stutter,” ICASSP, 2021
>
> [3] Zhou, Xuanru, et al. "Yolo-stutter: End-to-end region-wise speech dysfluency detection." Interspeech, 2024
>
> [4] Yamagishi, Junichi, et al. "CSTR VCTK Corpus: English multi-speaker corpus for CSTR voice cloning toolkit (version 0.92)."
>
> [5] Ardila, Rosana, et al. "Common voice: A massively-multilingual speech corpus." arXiv preprint arXiv:1912.06670 (2019).
>
> [6] Wang, Wenbin, et al. "GLOBE: A High-quality English Corpus with Global Accents for Zero-shot Speaker Adaptive Text-to-Speech." arXiv preprint arXiv:2406.14875 (2024).

---

> ### Author Response · Authors · 2024-11-19
>
> 4. Lack of Ablation Studies
>
> Thank you for raising this point!
> >
> > While we acknowledge the need for more ablation studies, we'd like to first clarify a potential misunderstanding. Table 1 focuses solely on evaluating the NAF (Neural Articulatory Flow) component, specifically assessing the intelligibility of our proposed gestural score representations. **This evaluation is independent of other modules like FCSA and NICL, as NAF can be trained separately**. In response to reviewer wcgM's request, **we have added WavLM-Large to our ablation studies and updated Table 1 (highlighted in blue)**. The results show that WavLM-Large provides only marginal improvements over HuBERT and still performs significantly worse than our NAF in phoneme recognition.
> >
> > **We assume you might have concerns for Table 3 (But please correct us)**, which requires a more comprehensive exploration of ablations across different modules (NAF, FCSA, NICL) and loss objectives in Eq. 11. This aligns with reviewer wcgM's feedback, and we respond as follows:
> >
> >> SSDM 2.0 has introduced three major modules (NAF, FCSA, NICL) and multiple loss objectives for each module, making it necessary to explore the importance of each component. While Table 3 discussed the results when replacing individual SSDM  modules, here we present additional ablation studies. For simplicity, we focus on Libri-Dys inference experiments.
> We first explain the notation. Starting with the baseline SSDM, single-module replacements are denoted as: _SSDM+NAF_, where we replace SSDM's gestural scores with our NAF gestural scores; _SSDM+FCSA_, where we replace SSDM's CSA with our FCSA; and _SSDM+NICL_, where we replace SSDM's vanilla language modeling with our NICL. We can also replace multiple modules simultaneously: _SSDM+NAF+FCSA_, _SSDM+NAF+NICL_, and _SSDM+FCSA+NICL_. Note that SSDM 2.0 is equivalent to _SSDM+NAF+FCSA+NICL_.
> >
> >> For loss objective ablations, we refer to the complete loss function in Eq. 11. The NAF module involves three losses:   $\lambda_1\mathcal{L}\_{\text{KL}}$ + $\lambda_2\mathcal{L}\_{\text{FLOW}}$ + $\lambda_6\hat{\mathcal{L}}\_{\text{PIT}}$, where only $\hat{\mathcal{L}}\_{\text{PIT}}$ can be ablated as the first two are essential. FCSA includes two losses: $\lambda_3\mathcal{L}\_{\text{PRE}}$ + $\lambda_4\mathcal{L}\_{\text{POST}}$, each of which can be ablated. NICL has a single loss $\lambda_5\mathcal{L}\_{\text{CON}}$ that can be ablated. These ablations are denoted as $\text{SSDM+NAF-}\hat{\mathcal{L}}\_{\text{PIT}}$, $\text{SSDM+FCSA-}\mathcal{L}\_{\text{PRE}}$, $\text{SSDM+FCSA-}\mathcal{L}\_{\text{POST}}$, and $\text{SSDM+NICL-}\mathcal{L}\_{\text{CON}}$.
> >
> > **All ablation results are presented in Table 12 in the updated manuscript**. In terms of both F1 score and Matching Score (MS), replacing any single module in SSDM leads to performance improvement. Replacing an additional module (two modules in total) further enhances performance. Regarding the loss function, the posterior interpretable training (PIT) loss appears to have minimal influence. An interesting observation with FCSA is that incorporating both losses, $\mathcal{L}\_{\text{PRE}}$ and $\mathcal{L}\_{\text{POST}}$, delivers strong performance. However, removing either one results in a performance drop, although this trend becomes less pronounced with more data available. **Overall, each module and each loss in our proposed framework demonstrates its effectiveness**. For scalability, when all components are integrated, scalability increases dramatically. However, using only one or two modules yields less effective scalability improvements.
> >
> **Please feel free to check Appendix A.18 which was highlighted in Blue in the update manuscript.**
> >
> *  Lastly, in Section 6.7, **we present results without the NICL module in Table 7 (highlighted by green)**. While these results show slightly lower performance compared to the full SSDM 2.0 model, they still significantly outperform SSDM, demonstrating the importance of the NICL module.

---

> > ### Comment · Reviewer_f1SE · 2024-11-27
> >
> > Thank you for addressing the comments.
> > The presentation of the paper is improved and I am editing the presentation score to 3. The experimental setup is more solid and no further clarifications required from my side. However, overall the contribution of this paper still appears to be incremental on top of SSDM and the entire framework focuses on a very niche task, which is considered out-of-domain compared with other general speech models such as LTU-AS, SALMONN, etc. I am still not convinced that the contribution is broadly significant to the ICLR community. Therefore I am holding on to the overall rating. With that said, I appreciate the authors' prompt response and effort to keep polishing their work.

---

> ### Author Response · Authors · 2024-11-27
>
> Dear Reviewer:
>
> We appreciate that **the original weaknesses and questions have been addressed**, and thank you for your **new concerns**. However, we respectfully disagree with these new concerns and would like to clarify:
>
> 1. This is not incremental work on SSDM. Our proposed models - Neural Articulatory Flow (NAF), Full Connectionist Subsequence Aligner (FCSA), and Non-fluency In-Context Learning (NICL) - are completely different from SSDM. If this is your primary concern for rejection, please specify which modules you consider incremental. We are happy to discuss more. **This also seems to contradict your original review's "strengths" section where you praised our contributions**.
>
> 2. Regarding "broad significance to the ICLR community": Reviewer cH6N raised a similar concern, suggesting we should focus on general "articulation-based dysfluency" rather than just nfvPPA. We believe **this concern has been addressed** as it relates to writing problems. Our SSDM 2.0 has significant applications in speech disorders, language screening, and language learning, as noted in your "strengths" section. Per ICLR's code of ethics (https://iclr.cc/public/CodeOfEthics), particularly "**Respect the Work Required to Produce New Ideas and Artefacts**" and "**Be Fair and Take Action not to Discriminate**," we disagree that our paper has limited ICLR audience appeal or would be discriminated. Moreover, SSDM was published at NeurIPS, which has a comparable audience size to ICLR.
>
> While **we're not suggesting any violation of ICLR's code of ethics from the reviewer**, we don't believe these are valid grounds for rejection. We welcome further clarification of your concerns and would appreciate it a lot if you present all concerns at once to allow for prompt rebuttals.

---

### Official Review · Reviewer_wcgM · 2024-11-04

**Soundness:** 3
**Presentation:** 3
**Contribution:** 3
**Rating:** 6
**Confidence:** 2

**Summary:**

The paper presents a novel approach to transcribing speech that includes non-fluencies (such as repetitions, blocks, and sound replacements) beyond traditional text transcription. The authors introduce SSDM 2.0, which improves upon previous models by offering a highly scalable speech representation system called Neural Articulatory Flow, a comprehensive Full-Stack Connectionist Subsequence Aligner (FCSA) for aligning dysfluent speech with text, and a new method for in-context pronunciation learning that enhances dysfluency detection. Additionally, they open-source the largest co-dysfluency corpus, Libri-Co-Dys, for future research. SSDM 2.0 outperforms existing dysfluency transcription models while reducing training complexity significantly. The paper demonstrates the scalability and efficiency of the proposed system in both single and co-dysfluency transcription tasks.

**Strengths:**

1. The paper introduces SSDM 2.0, a novel model that goes beyond traditional transcription to accurately capture non-fluencies, filling a gap in existing speech transcription research. It introduces creative components like Neural Articulatory Flow for scalable speech representation and the Full-Stack Connectionist Subsequence Aligner for precise dysfluency alignment.
2. SSDM 2.0 is thoroughly evaluated with rigorous experimentation across multiple datasets.
3. The paper clearly defines its focus on time-accurate dysfluency transcription and outlines the shortcomings of previous models.
4. The model’s ability to transcribe co-dysfluencies has broad implications for fields such as speech therapy, language learning, and automatic speech recognition.

**Weaknesses:**

The paper includes a great deal of detail. However, there are several aspects that could be further clarified:
1. The evaluation details of the HuBERT and Soundstorm models in Table 1 are not explained in the paper. For instance, the original HuBERT paper only tested on the LibriSpeech and LibriLight datasets. How was the HuBERT model in this paper trained, and what training data was used?
2. The Acoustic Encoder of the model is WavLM, but Table 1 only shows results for HuBERT. Since WavLM generally outperforms HuBERT, the authors should consider adding more experiments comparing WavLM with NAF.
3. While the paper introduces many new modules, the comparison of baseline models could be more diverse. For example, the authors could include results from directly using a speech encoder (e.g., HuBERT, Whisper encoder) to extract speech representations as input for the LLM, followed by fine-tuning with LoRA. This would provide a clearer understanding of whether the added modules effectively improve the model's performance.
4. Although many new modules are proposed, there is a lack of comprehensive ablation studies to verify whether each module is necessary. For example, Equation 11 includes six different loss terms, but there are no corresponding experiments demonstrating whether each loss term contributes to performance improvement. While some related experiments are included later in the paper, they remain incomplete.

**Questions:**

1. How was the HuBERT model trained in this paper, and what specific training data was used, given that the original HuBERT paper only evaluated on LibriSpeech and LibriLight datasets?
2. Why does Table 1 only show results for HuBERT when the Acoustic Encoder is WavLM, and can the authors provide additional comparisons between WavLM and NAF, considering WavLM generally outperforms HuBERT?
3. Could the authors include more diverse baseline model comparisons, such as using a speech encoder like HuBERT or Whisper encoder with fine-tuning via LoRA, to better assess the impact of the new modules on performance improvement?
4. Can the authors provide more comprehensive ablation studies to verify the necessity of each proposed module, including experiments showing the impact of each loss term in Equation 11 on model performance?

---

> ### Author Response · Authors · 2024-11-19
>
> We greatly appreciate the reviewers' time and their in-depth feedback on our paper.
>
> 1. _HubERT Training and Table 1 ablations._
> >
> > In Section 6.3, we would like to provide a more detailed explanation of our approach. We use HuBERT-Large representations as input and employ a two-linear probe ((D,D)+(D,40)) for **frame-level phoneme classification**, where D is the input dimension (768 for HuBERT-Large). **The phoneme labels in our task are dysfluent labels**. For example, the word "please" with its standard phoneme sequence "P P L L IY IY IY IY Z Z" could have dysfluent labels like "P P L L IY IY IY IY L L IY IY IY IY SIL SIL SIL SIL Z Z", indicating a repetition of "L IY" and a block.
> >
> > Due to the dysfluent nature of the labels, we found that traditional speech representations (HuBERT, WavLM, or Codecs) exhibit poor scalability: increasing the training data size and model capacity does not significantly improve phoneme classification performance. We observed some improvement by adding a CTC branch specifically for dysfluent phoneme recognition (without duration). Our current architecture therefore consists of two branches: a two-linear layer probe for phoneme classification and a two-layer CTC for phoneme recognition. During inference, we only utilize the first branch.
> >
> > Throughout this stage, we maintain either HuBERT or our NAF fixed. Our experiments with finetuning either HuBERT or NAF showed minimal performance differences.
> >
> >* WavLM ablation
> >
> >> **We have expanded our ablation studies to include WavLM-Large and updated Table 1 (highlighted in blue).** The results demonstrate that WavLM-Large offers only marginal improvements over HuBERT and still performs substantially below our NAF model in phoneme recognition. We welcome any feedback on our updated manuscript.
> >
>
> 2. _Could the authors include more diverse baseline model comparisons, such as using a speech encoder like HuBERT or Whisper encoder with fine-tuning via LoRA, to better assess the impact of the new modules on performance improvement?_
> >
> > Thank you for raising this point. Using a Whisper encoder with LoRA finetuning is the approach implemented in SALMONN [1] for spoken language understanding. We evaluated SALMONN-13B in zero-shot settings and **also fine-tuned it with our Libri-Co-Dys corpus (denoted as SALMONN-13B-FT in Section 6.5, Table 5)**. The results indicate poor performance, suggesting that this is not merely a data-driven problem. This further validates the effectiveness of our proposed modules.
> >
> [1] Tang, Changli, et al. "SALMONN: Towards Generic Hearing Abilities for Large Language Models." ICLR, 2024

---

> ### Author Response · Authors · 2024-11-19
>
> 3. _Can the authors provide more comprehensive ablation studies to verify the necessity of each proposed module, including experiments showing the impact of each loss term in Equation 11 on model performance?_
> >
> > This is a great suggestion which is also a request from Reviewer f1SE and thus we use the same reply.
> >
> > We conducted additional comprehensive exploration of ablations across different modules (NAF, FCSA, NICL) and loss objectives in Eq. 11. This aligns with reviewer wcgM's feedback, and we respond as follows:
> >
> >> SSDM 2.0 has introduced three major modules (NAF, FCSA, NICL) and multiple loss objectives for each module, making it necessary to explore the importance of each component. While Table 3 discussed the results when replacing individual SSDM  modules, here we present additional ablation studies. For simplicity, we focus on Libri-Dys inference experiments.
> We first explain the notation. Starting with the baseline SSDM, single-module replacements are denoted as: _SSDM+NAF_, where we replace SSDM's gestural scores with our NAF gestural scores; _SSDM+FCSA_, where we replace SSDM's CSA with our FCSA; and _SSDM+NICL_, where we replace SSDM's vanilla language modeling with our NICL. We can also replace multiple modules simultaneously: _SSDM+NAF+FCSA_, _SSDM+NAF+NICL_, and _SSDM+FCSA+NICL_. Note that SSDM 2.0 is equivalent to _SSDM+NAF+FCSA+NICL_.
> >
> >> For loss objective ablations, we refer to the complete loss function in Eq. 11. The NAF module involves three losses:   $\lambda_1\mathcal{L}\_{\text{KL}}$ + $\lambda_2\mathcal{L}\_{\text{FLOW}}$ + $\lambda_6\hat{\mathcal{L}}\_{\text{PIT}}$, where only $\hat{\mathcal{L}}\_{\text{PIT}}$ can be ablated as the first two are essential. FCSA includes two losses: $\lambda_3\mathcal{L}\_{\text{PRE}}$ + $\lambda_4\mathcal{L}\_{\text{POST}}$, each of which can be ablated. NICL has a single loss $\lambda_5\mathcal{L}\_{\text{CON}}$ that can be ablated. These ablations are denoted as $\text{SSDM+NAF-}\hat{\mathcal{L}}\_{\text{PIT}}$, $\text{SSDM+FCSA-}\mathcal{L}\_{\text{PRE}}$, $\text{SSDM+FCSA-}\mathcal{L}\_{\text{POST}}$, and $\text{SSDM+NICL-}\mathcal{L}\_{\text{CON}}$.
> >
> > **All results are presented in Table 12 in the updated manuscript**. In terms of both F1 score and Matching Score (MS), replacing any single module in SSDM leads to performance improvement. Replacing an additional module (two modules in total) further enhances performance. Regarding the loss function, the posterior interpretable training (PIT) loss appears to have minimal influence. An interesting observation with FCSA is that incorporating both losses, $\mathcal{L}\_{\text{PRE}}$ and $\mathcal{L}\_{\text{POST}}$, delivers strong performance. However, removing either one results in a performance drop, although this trend becomes less pronounced with more data available. Overall, each module and each loss in our proposed framework demonstrates its effectiveness. For scalability, when all components are integrated, scalability increases dramatically. However, using only one or two modules yields less effective scalability improvements.
>
> **Please feel free to check Appendix A.18 which was highlighted in blue in the update manuscript.**

---

> ### Author Response · Authors · 2024-11-27
>
> Dear Reviewer,
>
> Thank you for your detailed reviews, which have helped us significantly improve our manuscript. We would greatly appreciate your feedback on whether our revisions have adequately addressed your concerns. If so, according to recent PC regulations, we would like to sincerely request consideration for a score raising. If any concerns remain, we are happy to provide additional clarification or engage in further discussion.
>
> Thank you again for your time and consideration!
>
> Sincerely

---

### Official Review · Reviewer_i1kS · 2024-11-05

**Soundness:** 4
**Presentation:** 2
**Contribution:** 3
**Rating:** 6
**Confidence:** 4

**Summary:**

In this work, the authors tackle the problem of fine-grained speech transcription along with non-fluencies (dysfluencies). Towards this, they present a new system SSDM 2.0 (that builds on top of SSDM, Lian et al. 2024) with some key modifications including a new articulatory feature-driven flow-based model, a full-stack connectionist subsequence aligner for improved coverage of dysfluencies and mispronunciation in-context learning using LLMs. This system consistently outperforms SSDM on multiple evaluation benchmarks.

**Strengths:**

* The problem is well-motivated and the authors propose multiple new and innovative techniques (SSDM 2.0) to tackle the problem of time-accurate transcription of speech with dysfluencies.
* The authors open-source a large dysfluency corpus (Libri-Co-Dys) to encourage further work in this domain.
* SSDM 2.0 is significantly more performant (both in terms of accuracy and efficiency) compared to prior state-of-the-art (SSDM).

**Weaknesses:**

* The writing could be improved overall, especially the main technical parts in Sections 3, 4.
  * Rather than diving right into the details of each module in Section 3, it would have been useful for the reader to first get a complete description of the full SSDM 2.0 framework and the role of each module (neural articulatory flow, full-stack CSA, etc.) in the overall pipeline before learning about individual details.
  * Notation can be made cleaner in many places (e.g., inconsistent definition of H with H \in R^{K \times T} in Section 3.1 and H \in R^{T \times K} in Section 3.2), and illustrative examples would be useful especially for full-stack CSA.
  * There are also many references to Lian et al. 2024 which disrupts the overall flow and makes it less self-contained.

**Questions:**

* The authors should motivate the gestural scores in H, right in Section 3.1 -- How is it rule-generated, what is K, why is it sparse and how does H relate to articulatory gestures? Since X is already articulatory data, how does the information in H differ from that in X?  More about H only appears scattered across Sections 3.2 and 3.3.

* Can the authors further motivate their choice of using a count encoder + index generator to identify active regions? One could devise alternate schemes like using discretized binary gates (using Gumbel softmax) for every element in X_i to denote whether or not it is part of an active region. It would be useful to know if the proposed design was motivated by some articulatory or speech production priors.

* From the proposed SSDM 2.0 architecture and the demo page, the LLM responses are descriptive and in natural language. Apart from objective measures of F1/WPER, did the authors consider evaluating these free-form responses containing pronunciation feedback via a human evaluation?

* Libri-Co-Dys (with multiple dysfluencies) is presumably harder than Libri-Dys (with a single dysfluency). However, the F1 scores using SSDM 2.0 on Libri-Dys appear to be no worse than that on Libri-Co-Dys (across 30, 60, 100 training data settings). Could the authors comment on this?

* Is there any fraction of Libri-Co-Dys that does not contain dysfluencies? If not, it is useful to check how SSDM 2.0 performs on fluent instances. Please comment.

* Minor comment: There are no arrows coming out of 3 and 4 (inside yellow circles) in Figure 4 (under Pre-Alignment Training).

* Section 6.8 could be moved entirely to Appendix A.5.

* The authors should do a careful editorial pass of the draft. Listing some examples of typos below:
  * ARTCULATORY --> ARTICULATORY (Title of Section 3.2)
  * Traing Data --> Training data (Section 6.2)
  * Title of Section 6.4 does not parse

---

> ### Author Response · Authors · 2024-11-19
>
> We greatly appreciate your detailed reviews. Here are our responses:
>
> 1. _The writing could be improved overall, especially the main technical parts in Sections 3, 4._
>
> > We acknowledge that some descriptions were unclear. To address this, we have added Section 2 (highlighted in red) in the updated manuscript to provide an overview of SSDM 2.0.
> >
> >> SSDM 2.0 (shown in Fig. 2) processes dysfluent speech and a textual prompt as inputs, generating pronunciation transcriptions as output. To achieve this, we propose the _Neural Articulatory Flow (NAF)_, which generates scalable speech representations referred to as _gestural scores_. Subsequently, the gestural scores are aligned with the reference text using a _Full-stack Connectionist Subsequence Aligner_, producing aligned speech embeddings for each token in the reference text. These aligned embeddings, combined with pre-defined prompts, are then input into a LLaMA module for instruction tuning, a process we term _Non-fluency In-context Learning_. The following subsections provide a detailed explanation of each module. Please review Section 2 in our updated manuscript. Note that we have moved the "Related Work" section to the appendix to save space. However, we welcome your feedback on these changes.
> >
> > We also corrected the notations as pointed out. **Please check our updated manuscript for content highlighted by red.**
>
> 2. _Explanation on Gestural scores H, Articulatory trajectory data X and number of gestures K._
>
> > X is articulatory data in $R^{D\times T}$, where T represents time and D is the channel dimension. In our setting, D equals 12, corresponding to x,y coordinates of 6 articulators. As raw data, X is at a relatively low level, making it difficult to capture high-level information. Therefore, we use convolutional matrix factorization (also called dictionary learning) to decompose X into two parts: the dictionary G in $R^{T'\times 12 \times K}$, which contains K kernels, where each kernel is a basic time-series pattern in $R^{T'\times 12}$, and **the gestural scores H, which are the projections of X onto the kernel space G**. H has shape $R^{K\times T}$, where **K is the number of kernels (a hyper-parameter)**.
> >
> > **We visualized X, G, and H in Appendix A.5**. Essentially, H represents when different kernels are activated. To illustrate with an example from Appendix A.5, consider articulatory data X where only the upper lip (purple lines), lower lip (gray lines), and tongue dorsum (red lines) move. In the simplest case, kernel 1 (G1), kernel 2 (G2), and kernel 3 (G3) represent the basic moving patterns of the upper lip, lower lip, and tongue dorsum, respectively. The gestural score H then shows when each kernel is activated and for how long. In the example, H shows that kernel 1 is activated for 3 time steps, kernel 2 for 6 time steps, and kernel 3 for 3 time steps.
> This figure presents a simplified example for illustration; in reality, the patterns can be more complex, with one kernel potentially involving multiple articulators. In summary, **H provides a high-level abstraction of X that is explainable, sparse, and scalable** (as demonstrated in our experiments). This is analogous to speech SSL, where SSL units provide another form of high-level abstraction (in terms of semantics) of raw audio waves.
> >
> > Although we provide a detailed illustration in the Appendix, **we have added further explanation in Section 3.1 (highlighted in red) to enhance clarity, following your suggestion.**
>
> 3. _How is H Rule-Generated and How is Speech Production Prior Incorporated?_
>
> > Traditional speech representation is a $R^{D\times T}$ matrix, where values are densely learned through data-driven methods. Our intuition is that in human speech production (prior knowledge), only several articulators are moving, and the activation (gestural scores) of these articulators is really sparse (in terms of space and time). **This motivates us to ask: Can we develop rules to generate gestural scores instead of using traditional data-driven speech representation learning methods? By rules, we mean we simply predict the index of each activation for articulators and inject sparsity into them (restricted in the count encoder)**. So instead of predicting a $R^{D \times T}$ matrix as speech representation, we predict where and when the articulators are activated, which we term as rule-based or semi-implicit representation learning. Actually, this is not a purely rule-based system; it is more like representation learning with injected human priors.
> >
> > **Also following Reviewer f1SE's suggestion, we have added introductory sections to Section 3 (before 3.1) and Section 3.1 in our updated manuscript, highlighted in green.**

---

> ### Author Response · Authors · 2024-11-19
>
> 4. _Can the authors further motivate their choice of using a count encoder + index generator to identify active regions? One could devise alternate schemes like using discretized binary gates (using Gumbel softmax) for every element in $X_i$ to denote whether or not it is part of an active region. It would be useful to know if the proposed design was motivated by some articulatory or speech production priors.
> There are two reasons why we do not use element-wise prediction (whether or not it is part of an active region)._
>
> > First, as we illustrated before, based on human speech production priors, the human speech production system only cares about when each articulator is activated and for how long. This motivates us to develop a more compositional method to model the human speech production system, distinct from traditional element-wise dense speech representation learning methods.
> >
> > Second, **in the SSDM work, they already used such dense element-wise prediction**, predicting the intensity of each element followed by post-processing sparsity methods. That pipeline is overly complex, and we aim to simplify it. Additionally, **our region-wise prediction is much more efficient than element-wise prediction**. As shown in the example in Appendix A.5, element-wise prediction requires $K\times T$ predictions, while our index-count pipeline needs only 3 count predictions, 6 index predictions, and some intensity predictions. Note that in SSDM, H is still a $K \times T$ matrix which remains dense, whereas in our method, H is a sparse matrix and we only store the activated elements (about only 5%-10% of elements) during computation.
>
> 5. _From the proposed SSDM 2.0 architecture and the demo page, the LLM responses are descriptive and in natural language. Apart from objective measures of F1/WPER, did the authors consider evaluating these free-form responses containing pronunciation feedback via a human evaluation?_
>
> > That is a good suggestion. Our primary goal is to predict the time and type of dysfluencies, which is already well-evaluated by automatic metrics including F1 score and matching score (taking time IoU into consideration), as visualized in Appendix A.3.2. In terms of prediction correctness, human scoring may not be necessary. However, we identify two cases where human evaluation could be valuable:
> >
> >> First, we use pre-designed prompts to extract the time and type of dysfluencies. To verify that these prompts don't fail in certain cases (i.e., fail to correctly extract information), we can use human efforts to manually evaluate the correctness. This essentially evaluates the effectiveness of prompts as shown in Appendix A.11. We could use a scoring system of 0, 0.5, and 1, where 1 indicates the prompt successfully extracts both type and time, and 0 indicates failure.
> >
> >> Second, regarding interface design, users desire more friendly and effective feedback. Even when dysfluencies are detected, the feedback (such as "strengthen the pronunciation of sound k") is still generated somewhat randomly. We could use mean opinion scores, commonly used in TTS evaluation, to assess how satisfied humans are with the general feedback. We plan to implement these evaluations in the future. Additionally, whether such labeled data can further benefit training (like RLHF) might also be an interesting topic to explore.

---

> ### Author Response · Authors · 2024-11-19
>
> 6. _Libri-Co-Dys (with multiple dysfluencies) is presumably harder than Libri-Dys (with a single dysfluency). However, the F1 scores using SSDM 2.0 on Libri-Dys appear to be no worse than that on Libri-Co-Dys (across 30, 60, 100 training data settings). Could the authors comment on this?_
>
> > This is an interesting point. To clarify, you mean "F1 scores using SSDM 2.0 on Libri-Dys appear to be **worse** than those on Libri-Co-Dys" (versus your initial point about "no worse").
> >
> > There are two categories.
> >
> >> (1) Phonetic transcription results.
> >> Looking at results from Table 1 and Table 2, for Libri-Dys, the scores are 92.0, 93.2, 95.0, and for Libri-Dys-Co, they are 92.7, 93.8, 96.0 respectively. Although Libri-Co-Dys contains more complex speech patterns, the results seem to be consistently better than those on the seemingly easier Libri-Dys data.
> We believe this is due to the nature of the labels. The labels we used for training and evaluation in Table 3 and Table 4 are real dysfluent phoneme labels. For example, assume the original text is "P L IY Z". After simulation, for Libri-Dys, the label might be "P L EY Z" indicating a replacement, while for Libri-Co-Dys, the label might be "P L P L EY Z" indicating both a stutter and a replacement. If we were to use the original text "P L IY Z" as labels, we would expect lower F1 scores on Libri-Co-Dys since it contains more disrupted speech. However, since **we use the actual "dysfluent phoneme" transcriptions (what people actually said)**, the labels in both Libri-Co-DYs and Libri-Dys accurately reflect the speech content. Therefore, these are not actually "harder" samples. **Data becomes "harder" only when labels don't match the speech (such as when using fluent text as labels)**.
> >
> >> (2) Dysfluency detection
> Looking at results from Table 3 and Table 4, for Libri-Dys, the scores are 80.0, 83.2, 86.2, and for Libri-Dys-Co, they are 81.4, 83.0, and 87.0 respectively. Results on Libri-Co-Dys appear to be better. For this task, the labels are dysfluency labels, formatted as {Phoneme: "IY", Start: 0.68, End: 0.79, Type: "replace"}, as shown in Appendix A.2.1. **Since we have accurate dysfluent labels, Libri-Co-Dys samples are not necessarily harder**. Samples typically become harder when labels are less accurate or match the speech less well.
> Consider an analogy from image recognition: if an image shows a dog in the foreground and a cat in the background, and the label is just "dog," this would be a harder sample than an image containing only a dog. However, if the label is "dog + cat" perfectly matching the image content, it becomes difficult to determine whether this is a hard or easy sample. But this is indeed an interesting point.
> For both cases, as for why Libri-Dys-Co performs slightly better than Libri-Dys, we believe this is because Libri-Co-Dys is larger than Libri-Dys (6k versus 4k hours). Thus, for each portion (30%, 60%, 100%), we have more data in Libri-Co-Dys. We welcome any comments on this analysis.
>
> 7. _Is there any fraction of Libri-Co-Dys that does not contain dysfluencies? If not, it is useful to check how SSDM 2.0 performs on fluent instances. Please comment._
>
> > Thanks for bringing this up. Libri-Dys-Co does not contain fluent speech. Let's examine speech recognition from a unified perspective. Assume a person says "Pl-l-l-ease c-c-call stella"; a good ASR system should provide two outputs: "what the person actually said" (the pronunciation transcription) and "what the person intended to say" (which is what current ASR models like Whisper produce). Our system can accurately handle both tasks.
> >
> > For the latter output, if we simply ask the system "what the person intended to say," it functions as a normal ASR system, and we report these results in Table 8. Note that under this setting, the targets/labels are the ground truth text "what people should have said." Since SSDM 2.0 already demonstrates good performance in this task (even better than Whisper), we speculate that it will also perform well on fluent speech in terms of ASR.
> >
> > However, it is indeed important to test these two tasks on fluent speech. To do this, we run SSDM 2.0 on LibriTTS test-clean and test-other sets to conduct two experiments:
> >
> >> Dysfluency pronunciation transcription (we report false positives: how many pronunciation problems are detected when they don't actually exist)
> >
> >> Normal ASR where we prompt the system to generate what the person intended to say
> >
> > **We report both new results in Appendix A.15 in the updated manuscript**. The basic conclusions are that for the dysfluency detection task, we found only a small portion of false positives, which indicates some room for improvement in future work. For the ASR task, performance is comparable but slightly worse than Whisper, which we attribute to not having tuned the model specifically for ASR tasks.

---

> ### Author Response · Authors · 2024-11-27
>
> Dear Reviewer,
>
> Thank you for your detailed reviews, which have helped us significantly improve our manuscript. We would greatly appreciate your feedback on whether our revisions have adequately addressed your concerns. If so, according to recent PC regulations, we would like to sincerely request consideration for a score raising. If any concerns remain, we are happy to provide additional clarification or engage in further discussion.
>
> Thank you again for your time and consideration!
>
> Sincerely

---

### Author Response · Authors · 2024-11-19
**Global Response Regarding Paper Update**

Dear Reviewers:

> Thank you for your thorough and in-depth reviews of our paper. We have updated the manuscript by reorganizing sections and adding additional experiments. Below is a global response summarizing the changes made to address your reviews. Note that we only discuss reviews that led to updates in the paper; questions directly addressed in responses are not listed here.
>
> For clarity, we use different colors to highlight updates made according to each reviewer's feedback:
>
>> Reviewer cH6N: yellow
>
>> Reviewer f1SE: green
>
>> Reviewer wcgM: blue
>
>> Reviewer i1kS: red
>
> When multiple reviewers raised the same concern, we randomly selected one color to represent the change.
> Summary of changes in the updated manuscript:
>
>> (1) Enhanced clarity regarding dysfluency and clinical background in the introduction (highlighted in yellow, addressing Reviewer cH6N's concerns)
>
>> (2) Improved and simplified introduction to articulatory modeling in Sections 3 and 3.1 (highlighted in green and red, addressing concerns from Reviewers cH6N, f1SE, and i1kS)
>
>> (3) Added new Section 2 summarizing the SSDM 2.0 pipeline (highlighted in red, addressing concerns from Reviewers f1SE and i1kS)
>
>> (4) Added experiments on accented speech (VCTK, Common Voice, GLOBE), addressing Reviewer cH6N's major question and Reviewer f1SE's concern about real-world data experiments (highlighted in yellow, results in Appendix A.16 and Table 10)
>
>> (5) Tested SSDM 2.0 on real-world stuttered speech datasets UCLASS and SEP28K (highlighted in green, addressing Reviewer f1SE's concerns, results in Appendix A.17 and Table 11)
>
>> (6) Added extensive ablation studies across modules (NAF, FCSA, NICL) and loss objectives (addressing questions from Reviewers wcgM and f1SE, results in Appendix A.18 and Table 12)
>
>> (7) Added WavLM ablation in Table 1 (highlighted in blue, as suggested by Reviewer wcgM)
>
>> (8) Added w/o NICL ablation in Section 6.7 (highlighted in green, as suggested by Reviewer f1SE)
>
>> (9) Added fluent speech (LibriTTS) dysfluency ablation to explore false positives (highlighted in red, as suggested by Reviewer i1kS, results in Appendix A.15 and Table 9)
>
>> (10) Relocated Related Work section to Appendix B and adjusted image and table sizes to optimize space
>
>> (11) Corrected typos as indicated by specific reviewers (marked in corresponding colors)
>
>  We are happy to address any additional concerns or remaining questions that reviewers may have. Please feel free to reach out for further clarification.

---

### Author Response · Authors · 2024-11-21
**Global Response Regarding the Second Update of the Paper**

Dear Reviewers,

We have made a second update to our paper as suggested by reviewer cH6N, with the changes highlighted in deeper yellow. Here are the updates:
>
> * In Introduction, we updated the first two paragraphs again by constraining our focus to pathological speech disorders only, specifically articulatory-based dysfluencies. The abstract and conclusion have also been slightly edited.
>
> * In Section 3, we stated that articulatory representation learning is the natural way for articulatory dysfluency modeling.
>
> * We removed codec experiments from Table 1 to reduce confusion.
>
There are no updates regarding other reviewers' reviews since the last update. We really appreciate the  in-depth suggestions that we have received to make the manuscript more solid. Please feel free to propose any questions or suggestions!

---

### Author Response · Authors · 2024-12-02
**Summary of Reviews from Authors**

Dear reviewers,

Thank you for your valuable reviews. As we have not received further discussion regarding our rebuttal, **we believe this indicates that all concerns and weaknesses raised by reviewers have been adequately addressed**. Given that we have addressed all concerns, we would greatly appreciate your consideration in raising the scores accordingly. We remain available to provide any additional clarification should you have new suggestions.

Sincerely

---

### Note · Authors · 2025-01-22

**Comment:**

Thank you for all reviewers' and AC's comments. While we believe we have thoroughly addressed both the writing and experimental concerns through extensive revisions in the updated manuscript, it seems these changes were not fully considered in the AC's final comments. We would like to address these points:

* Regarding the AC's concern about the introduction, we followed reviewer cH6N's "valuable suggestion" in revising it. Interestingly, this writing issue may not have existed in our initial draft. **This creates a paradox: our initial draft had a clearer story → reviewer cH6N suggested revisions → we implemented these changes to reviewer cH6N's satisfaction → AC questioned these changes while simultaneously endorsing the reviewer's suggestions**. This has left us confused about the best path forward. The presentation issues raised by other reviewers appear to have been adequately addressed, as we received no further feedback on these points. Regarding the statement "The gap between lexicalized speech and uttered speech is referred to as dysfluency," this was presented in the context of disordered speech, as suggested by reviewer cH6N. We would welcome more specific feedback to improve its clarity.

* Concerning the "lack of controlled experiments and ablation studies" raised by reviewers wcgM and f1SE: While we acknowledge that some ablations were initially missing, we have since conducted comprehensive experiments to address these gaps. However, we received no feedback on these additional results. At this point, we are unclear which specific ablations or controlled experiments are still considered missing, as our team believes we have completed all requested experimental work.

* Regarding other concerns raised by the AC:

  * On the demo page showing "a different task, not disfluency detection": We would like to clarify that we include phonetic errors as a type of dysfluency, so the tasks are indeed related.
  * Concerning the misuse of terms like "flow" and "full-stack": We appreciate this feedback and would be happy to replace these terms with more precise terminology that better fits the academic context.
  * Regarding the comment "there are good reasons for reviewers not to raise their scores. Continuously asking the reviewers to raise their score is not a healthy attitude to improve the paper", we sincerely appreciate all the insightful reviews and have incorporated them to improve our manuscript. We apologize if our requests for score reconsideration came across as inappropriate. However, we find ourselves in a puzzling situation where reviewers acknowledge that issues have been addressed, yet maintain their original scores. For the benefit of improving our work, we would greatly appreciate understanding the specific "good reasons for reviewers not to raise the scores." A more transparent review process with clear feedback would be more constructive than one where authors must *guess* at unstated concerns.

In conclusion, **we are uncertain why these addressed concerns remain the primary reasons for rejection**. While we fully respect ICLR's decision and acknowledge there is nothing more we can do at this stage, we still appreciate the insightful reviews and suggestions provided by the reviewers, as noted by the AC. However, we would have appreciated a more technically focused review process. Given these circumstances and the final result, we have decided to withdraw the paper and we remain open to any constructive suggestions to improve our work!

**Withdrawal Confirmation:**

I have read and agree with the venue's withdrawal policy on behalf of myself and my co-authors.